# Copy-choice recombination during mitochondrial L-strand synthesis causes DNA deletions

Örjan Persson[1], Yazh Muthukumar[1], Swaraj Basu [1], Louise Jenninger[1], Jay P. Uhler[1], Anna-Karin Berglund[1], Robert McFarland[2], Robert W. Taylor[2], Claes M. Gustafsson[1], Erik Larsson[1] & Maria Falkenberg[1]

Mitochondrial DNA (mtDNA) deletions are associated with mitochondrial disease, and also accumulate during normal human ageing. The mechanisms underlying mtDNA deletions remain unknown although several models have been proposed. Here we use deep sequencing to characterize abundant mtDNA deletions in patients with mutations in mitochondrial DNA replication factors, and show that these have distinct directionality and repeat characteristics. Furthermore, we recreate the deletion formation process in vitro using only purified mitochondrial proteins and defined DNA templates. Based on our in vivo and in vitro findings, we conclude that mtDNA deletion formation involves copy-choice recombination during replication of the mtDNA light strand.

---

[1] Department of Medical Biochemistry and Cell Biology, University of Gothenburg, P.O. Box 440, Gothenburg SE-405 30, Sweden. [2] Wellcome Centre for Mitochondrial Research, Institute of Neuroscience, Newcastle University, Newcastle upon Tyne NE2 4HH, UK. These authors contributed equally: Örjan Persson, Yazh Muthukumar, Swaraj Basu. Correspondence and requests for materials should be addressed to E.L. (email: erik.larsson@gu.se) or to M.F. (email: maria.falkenberg@medkem.gu.se)

The circular human mitochondrial genome (mtDNA) compromises 16,569 base pairs (bp) and codes for 13 protein components of the oxidative phosphorylation machinery. Large-scale deletions of mtDNA are associated with a number of human mitochondrial diseases and also accumulate with human ageing in post-mitotic tissues[1]. Deletions may occur sporadically[2], e.g., chronic progressive external ophthalmoplegia (CPEO) and Kearns–Sayre syndrome (KSS)[3,4]. Alternatively, deletions may be caused by Mendelian-driven mutations in proteins involved in mtDNA maintenance, including the mtDNA replication machinery and enzymes maintaining mitochondrial nucleotide pools[5].

The majority of mtDNA deletions seen in mitochondrial disease and formed during ageing are located in the major arc between the two origins of mtDNA replication (OriH and OriL). The striking distribution of deletions suggests the existence of an underlying mechanistic preference[6,7]. The breakpoints bordering the more frequently observed deletions often have direct repeat sequences[6] and are identified in non-B-structure-forming areas of DNA, such as hairpin and G4 structure-forming regions[8–10]. Deletions are classified according to the presence or absence of flanking repeat sequences. Of the more than 800 clinically significant deletions reported, ~60% are defined as class I deletions, i.e., have occurred between homologous direct repeats. In another ~30% of cases, the deletions are formed between imperfect repeats (class II), whereas the remaining deletions are formed between non-repeated sequences (class III)[11]. The most frequently reported mtDNA deletion is the loss of a ~5.0 kb region between two direct repeats of 13 bp (ACCTCCCTCACCA), beginning at position 8470 and position 13447. In this so called "common deletion", one of the repeats is retained and the other lost together with the intervening sequence during the rearrangement process, leading to a deficiency of essential mitochondrial genes encoding both mRNAs and tRNAs. The common deletion was first detected in patients with mitochondrial myopathy[12], but was later shown to also accumulate during normal ageing in humans[1].

The mitochondrial replication machinery is distinct from that found in the nucleus and includes: DNA polymerase γ (POLγ), a heterotrimer consisting of one catalytic subunit POLγA and two subunits of the processivity factor POLγB; the replicative DNA helicase TWINKLE, required for POLγ-dependent replication on double-stranded DNA (dsDNA) templates; and the mitochondrial single-stranded DNA-binding protein (mtSSB), which further stimulates replication[13]. According to the strand-displacement model (SDM) for mtDNA replication, leading-strand DNA synthesis is initiated at the heavy-strand origin (OriH), which is located in the non-coding region (NCR) of the mtDNA molecule (Fig. 1a)[14,15]. From OriH, DNA synthesis proceeds unidirectionally to produce the nascent H-strand. When the replication machinery has reached about two-thirds of the genome, it passes the light-strand origin of replication (OriL). When OriL is exposed in its single-stranded conformation, it becomes available for primer formation and DNA synthesis can begin in the opposite direction[16]. Once initiated, both H-strand and L-strand synthesis proceed continuously until two new daughter molecules are formed, which are separated in a process involving the mitochondrial isoform of Topoisomerase 3α (TOP3A)[17].

The mechanisms underlying mtDNA deletion formation remain unclear, but two major mechanistic theories have been proposed. The slipped-strand model, first presented in 1989, suggests that the deletion process is initiated during H-strand DNA synthesis across the major arc, when the parental H-strand is single stranded[18]. A second model proposes that mtDNA deletions are caused by double strand breaks followed by DNA repair[19]. In the present report, we combine analysis of mtDNA deletions formed in vivo with mechanistic studies of deletion formation in vitro. Our data argue against the previously proposed models and instead suggest that mitochondrial deletions are formed by copy-choice recombination during active L-strand DNA synthesis.

## Results

**A distinct directionality of mtDNA deletions in vivo.** Mutations in POLγ (encoded by the *POLG* gene) cause multiple deletions in vivo. We first decided to analyse repeated sequence elements associated with deletion breakpoints in patients with inherited disorders of mtDNA maintenance. We isolated total muscle DNA from three adult patients with late-onset PEO caused by compound heterozygous *POLG* variants and performed deep DNA sequencing. Our analysis revealed extensive variable mtDNA deletions that were highly consistent in between the three samples (Fig. 1b). The vast majority of detected deletions were in the major arc, located in the region between OriH and OriL (Fig. 1a, b and shown as histograms of breakpoint positions in Supplementary Fig. 1). Deletions could be detected also in the healthy controls, which was as expected due to the higher age of these individuals (82 and 86 years). The levels of deletions in the patient samples were however much higher, even after correcting for variable sequencing depth (Supplementary Fig. 2). Similar to previous reports, we noticed that the majority of OriH-proximal breakpoints clustered near the end of the non-coding region. All identified and analysed deletions can be found in Supplementary Data 1.

Further analysis, comparing the specific sequence contexts near the ends of each deletion, revealed that the breakpoints were enriched for direct repeats. In particular, exact sequence matches of length 5 bp or above were observed much more frequently than expected compared with randomized breakpoints (1.8-fold for 5 bp; 8.3-fold for 8 bp; Fig. 1c). The observed repeats were enriched for nucleotide motifs that were rich in cytosines, and notably this was observed also for shorter matches (e.g., CCC, ACC, ACCC; Fig. 1d). A subset of the observed deletions (119 out of 623, 19.1%) showed imperfect repeats that could be used to address the directionality of the events, i.e., on what side of the deletion the repeat was retained or lost (Fig. 1e, Supplementary Data 1). Eighty and 39 of the informative cases (67.2% and 32.7%) supported retention of the repeat at the 5′ and 3′ side, respectively. These numbers improved to 122 and 40 (75.3% and 24.7%) when considering each supporting read individually rather than unique deletions, indicating a distinct directionality in the deletion process.

To investigate if the observed directionality was unique to mutations in *POLG*, we also analysed breakpoints in adult patients with late-onset PEO caused by compound heterozygous, recessive variants in the *TOP3A* gene[17] or a well-characterised, dominant p.Arg357Pro variant in the *TWNK* gene, encoding the TWINKLE DNA helicase[20]. The identified pattern of mtDNA deletions was similar to those associated with *POLG* mutations (Supplementary Figs 3 and 4 and Supplementary Data 1). We identified 399 unique deletions in the TOP3A patient, including 31 instances of the common deletion. In total, 69 (17.3%) of the identified deletions showed imperfect repeats. Of these, 42 (60.8%) supported retention of the repeat at the 5′ side (Supplementary Fig. 3a). Nucleotide motifs enriched at breakpoints were similar to those identified with mutant *POLG* (Supplementary Fig. 3b, c). We identified fewer mtDNA deletions (181 unique deletions) in the patient with mutated *TWNK* and the common deletion was identified in 11 instances. In total, 26 (14.4%) of the identified deletions showed imperfect repeats. Of

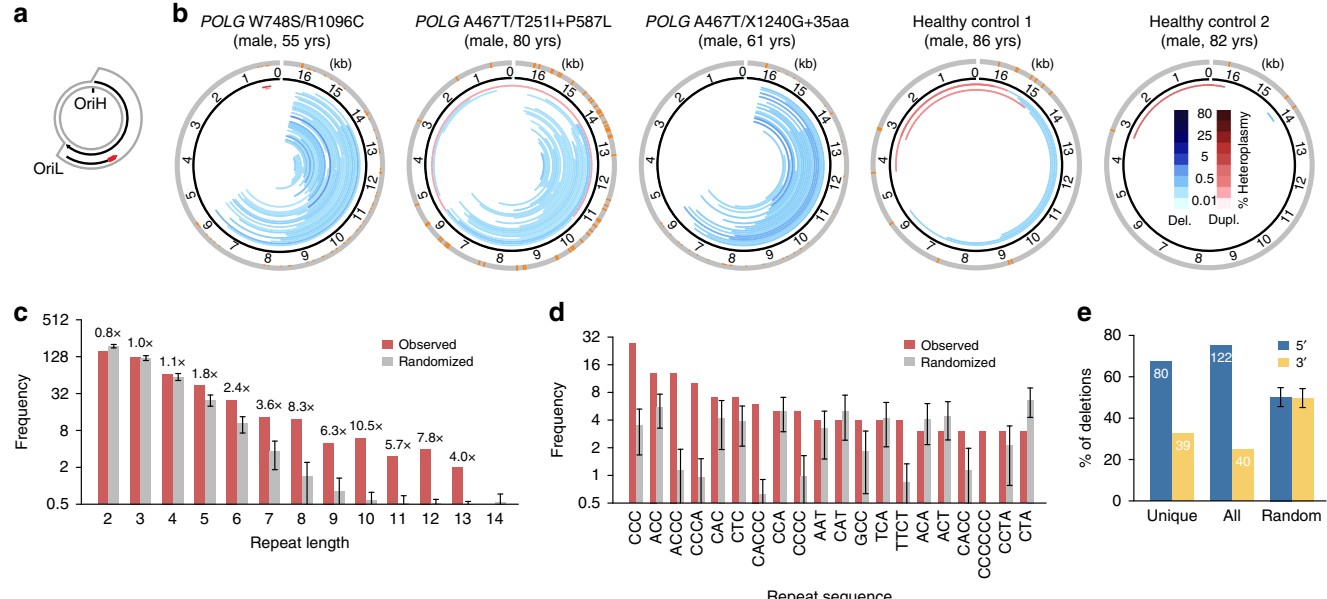

**Fig. 1** mtDNA deletions identified in patients with mutations in POLγ. **a** Strand-displacement DNA replication is continuous on both strands and initiated from two separate origins. **b** mtDNA deletions (blue) and duplications (red) predicted using high-throughput sequencing of skeletal muscle DNA from three patients with pathogenic, compound heterozygous *POLG* variants as well as two healthy control patients. The orange bars in the outermost circle show breakpoint frequency at a given base position. **c** Frequencies of exact direct repeats overlapping or flanking each pair of breakpoints, considering the longest match for each deletion and pooling the three patients (red bars). Randomized breakpoints are shown for comparison (grey bars), with error bars indicating the standard deviation (100 randomizations). **d** Frequencies for the most commonly observed repeat patterns. **e** Analysis of 5′ vs. 3′ retention of imperfect repeats. A subset of deletions where the breakpoint positions could be safely determined to within 1 bp were analyzed for the presence of imperfect direct repeats at both sides of the deleted segment and were classified as either 5′ or 3′ based on the longest discovered repeat having at most 1 mismatch. Results are shown for unique deletions as well as the complete set of deletions, pooled from the three *POLG* patients. Results from 100 sets of randomly generated deletions, each being similar in size as the observed data, are included. Error bars indicate the standard deviation

these, 15 (57.8%) supported retention of the repeat at the 5′-side (Supplementary Fig. 4a). Also, here the nucleotide motifs enriched at breakpoints were similar to those identified with mutant *POLG* (Supplementary Fig. 4b, c).

**A replication barrier promotes copy-choice recombination.** The directionality of the deletion events argues against the two previously proposed models for deletion formation. The slip-replication model would preferentially retain the repeat close to OriH, whereas the DNA repair model should take place without directionality. Instead, our findings pointed to copy-choice recombination, a mechanism observed in e.g., *E. coli*. According to this model, deletions are formed during lagging-strand DNA synthesis, when the template strand is single-stranded[21]. Single-stranded templates can form secondary structures that act as barriers for DNA synthesis, which in turn can cause replication pausing and DNA polymerase dissociation. As a consequence, the 3′-end of the nascent lagging strand may leave the template strand during active DNA synthesis and reanneal (mispair) to a downstream region. When DNA synthesis reinitiates from the reannealed 3′-end, the intervening sequence is deleted[22].

In the mitochondria, L-strand DNA synthesis is analogous to lagging-strand replication, using the single-stranded H-strand as a template. To further address if copy-choice recombination could explain deletion formation, we decided to reconstitute L-strand DNA synthesis and investigate the mechanism underlying the formation of mtDNA deletions. As a template, we used a circular, single-stranded DNA molecule of ~4300 nucleotides (Fig. 2a). The template included two ~600-bp regions of the mtDNA H-strand, each encompassing one of the two direct 13-bp repeats involved in common deletion formation, here designated 8470-CR and 13447-CR in reference to their genomic

loci (GenBank accession number: NC_012920.1). Since deletion formation is an extremely rare event, we also included a hairpin structure as an artificial replication barrier, which would impair POLγ progression (Fig. 2a and Supplementary Table 1, wild-type-barrier template, T1). This approach has been used in other systems to stimulate deletion formation to detectable levels with other DNA polymerases[22]. To prime L-strand synthesis, we annealed a radioactively labelled DNA oligonucleotide upstream of the direct repeat at 8470-CR (Fig. 2a, b). The hairpin replication barrier could indeed stall the mitochondrial replication machinery (Fig. 2a, c and Supplementary Fig. 5a). Addition of POLγ caused elongation of the primer, but the majority of replication events stalled after about 1500 nucleotides, at the location of the barrier. With time, POLγ could pass the barrier and increasing levels of longer products were formed (Fig. 2c). To determine if the passage involved copy-choice recombination, we performed PCR using a pair of primers flanking the inserted mitochondrial sequences (Fig. 2b). The input template and full-length daughter molecules, produced by POLγ progression through the hairpin region, generate a PCR product of 1500 bp. If copy-choice recombination occurs between the two 13-bp repeats, a heteroduplex molecule will be formed and the PCR reaction will generate an additional product of 750 bp. We observed the full-length 1500-bp PCR product in all reactions and, with time, we also observed the shorter 750-bp PCR product (Fig. 2d, lanes 5–7). We also noted the appearance of a shorter 350-bp PCR product (discussed below).

To demonstrate that the 750-bp fragment was not formed during the PCR analysis, we performed a control experiment in which the products formed during POLγ-dependent L-strand synthesis were cleaved with EcoRI. The recognition site of EcoRI lies in the hairpin loop, and EcoRI treatment will therefore

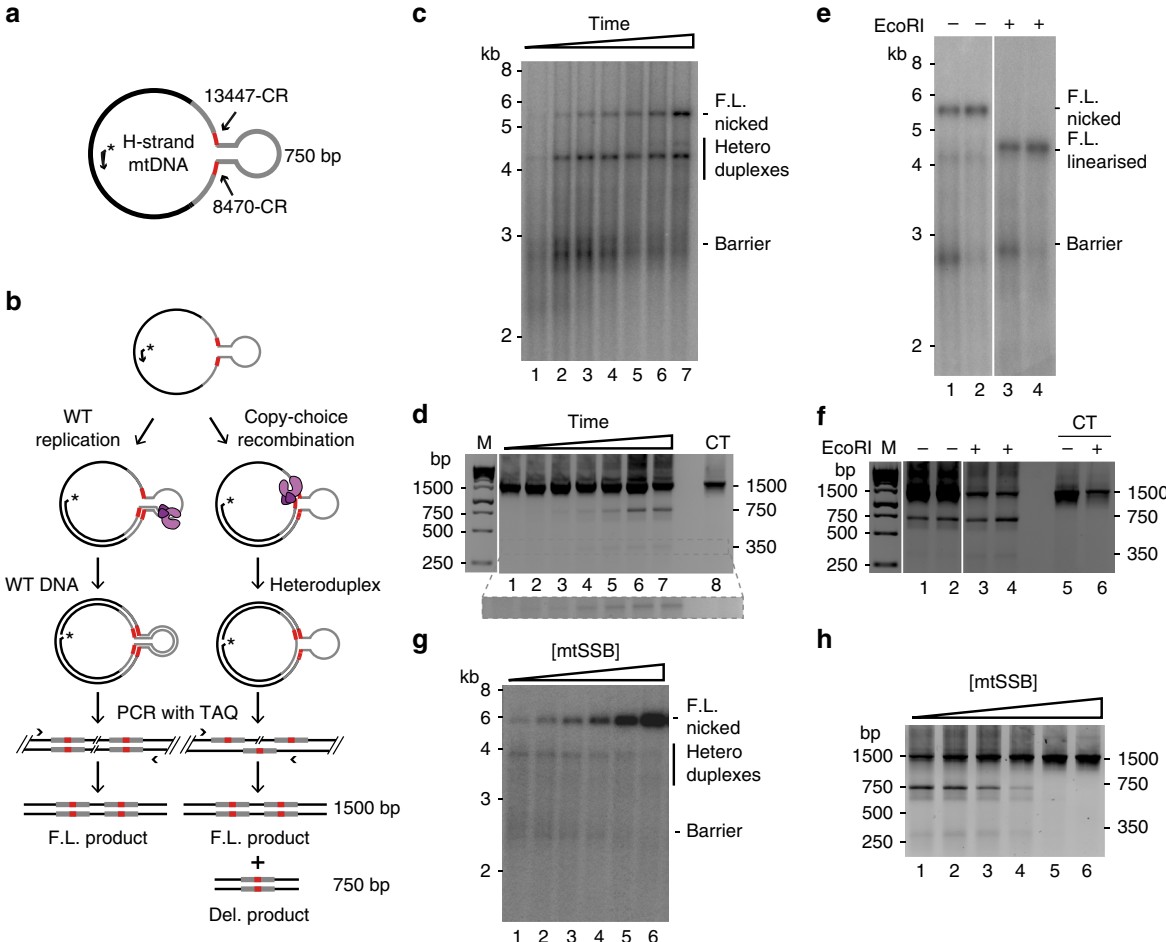

**Fig. 2** Deletion formation during L-strand synthesis. **a** Schematic representation of the barrier template (T1). The template contains two stretches of the mtDNA H-strand (each stretch ~600 bp, indicated in grey) including the 13-bp common repeats, 8470-CR and 13447-CR (indicated in red). A GC-rich inverted repeat is inserted at the 3′-end of 8470-CR and 5′-end of 13447-CR to form a hairpin structure that functions as a replication barrier. A [32]P-labelled primer for initiation of L-strand synthesis is annealed to the ssDNA template. **b** Schematic representation of the assay. POLγ (purple) initiates L-strand synthesis from a [32]P-labelled primer. If copy-choice recombination occurs during replication, a heteroduplex molecule is formed, containing one full-length H-strand and a deleted L-strand. The heteroduplex can be identified by PCR analysis, generating a shorter PCR product (750 bp) in addition to the full-length product of 1500 bp. **c** A time course (2.5, 5, 10, 15, 20, 30, 60 min) experiment of L-strand mtDNA synthesis was performed as described in Methods. The size markers are linearized dsDNA. **d** The reactions in panel c were analysed by PCR as described in panel b. The lower panel shows an expanded image of the boxed 350-bp region that has been further exposed. CT—control: single-stranded DNA template. **e** L-strand DNA synthesis incubated for 30 min (lanes 1 and 3) or 60 min (lanes 2 and 4) as in panel c, but before (lanes 1 and 2) and after EcoRI treatment (lanes 3 and 4). **f** The reactions in panel e were analysed by PCR. The double-stranded barrier plasmid before (lane 5) and after linearization with EcoRI (lane 6) was used as a control (CT). **g** L-strand DNA synthesis was performed as in panel c, but in the presence of increasing mtSSB concentrations (0, 0.5, 1.25, 2.5, 5, 12.5 nM calculated as tetramers). **h** The reactions in panel g analysed by PCR. Source data are provided as a Source Data file

linearize the full-length replication product but not the heteroduplex molecule (Fig. 2e and Supplementary Fig. 5b). Indeed, treatment with EcoRI before PCR analysis led to a specific reduction of the full-length 1500-bp products, without affecting the levels of the shorter 750-bp product (Fig. 2f).

Single-stranded DNA-binding proteins help to remove secondary structures in template DNA, thereby preventing replication stalling and deletion formation. When we repeated our experiments with increasing mtSSB concentrations, it led to the formation of more full-length products (Fig. 2g), and decreased deletion formation (Fig. 2h).

**Direct repeats facilitate deletion formation**. We next mutated the common repeat motifs to verify their role in deletion formation (Fig. 3a–c and Supplementary Table 1, templates T2–T4). Disruption of 13447-CR or both repeats (8470-CR and 13447-CR) prevented formation of the 750-bp product (Fig. 3c).

Surprisingly, when only the 8470-CR repeat was mutated, there was no effect on deletion formation (Fig. 3c, T2). In order to understand the precise nature of the deleted products, PCR products were cloned and sequenced. Sequencing of the 750-bp products produced with the wild-type-barrier template (Supplementary Table 2, T1 template- POLγ wild-type and Supplementary Fig. 6a, b, lanes 3 and 4) generated 164 different clones, all of which corresponded to deletions formed between direct repeats. In all cases, one of the repeats was lost together with the intervening sequence (Fig. 3d and Supplementary Table 2). However, only 26% of these were due to perfect copy-choice recombination between the two 13-bp repeats (8470-CR and 13447-CR) (Fig. 3d). The remaining 74% originated from copy-choice recombination between sequences within the stem-region and the 13447-CR sequence (Fig. 3d and Supplementary Table 2). Apparently, POLγ progresses a distance into the stem-region before dissociating. The 3′-end of the newly replicated strand

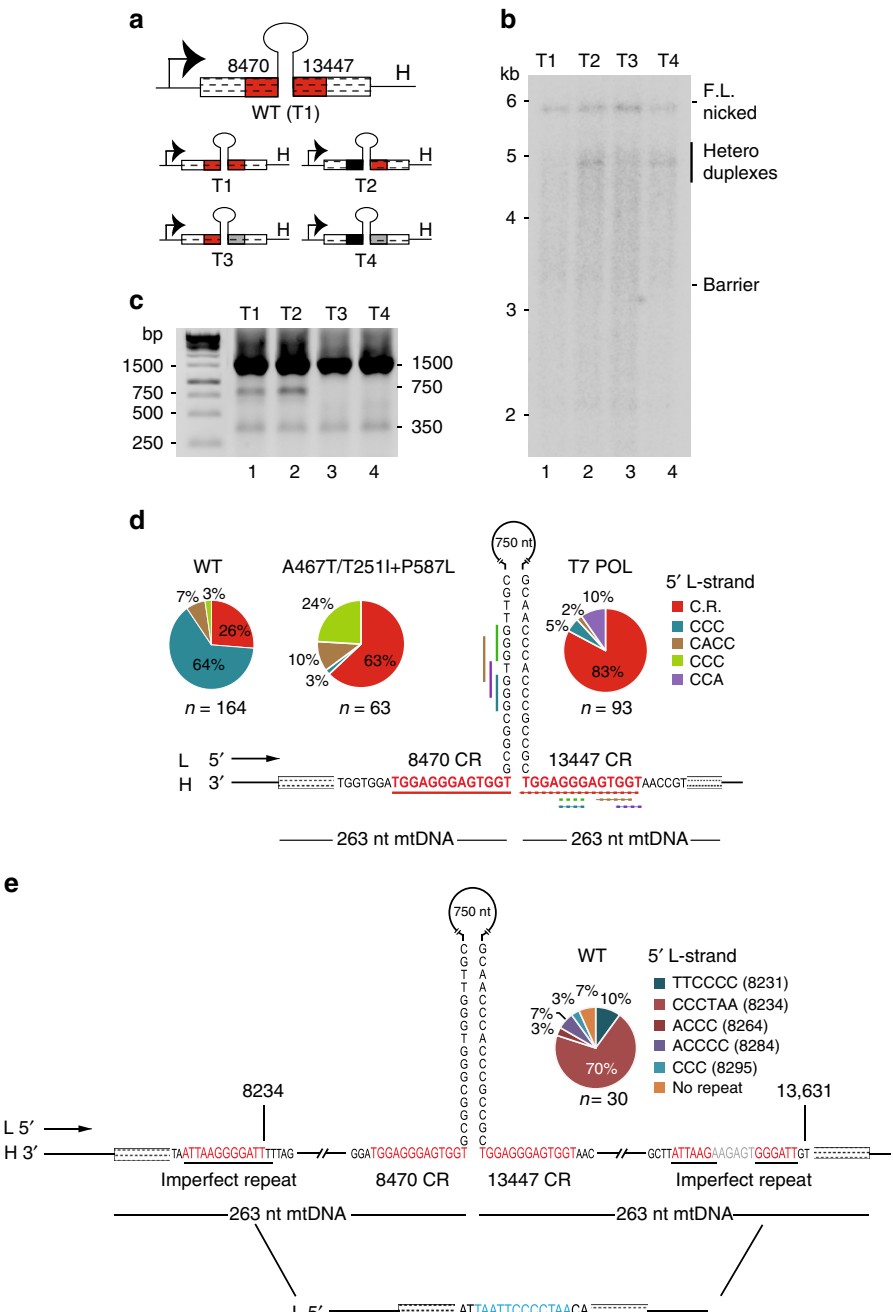

**Fig. 3** Deletion formation between direct repeats. **a** Mutation analysis of 8470-CR and 13447-CR. Barrier templates were used with wild-type repeats (T1); or with mutations in 8470-CR (T2); 13447-CR (T3); 8470-CR and 13447-CR (T4). **b** L-strand DNA synthesis using the T1-T4 templates. **c** The reactions in panel **b** analysed by PCR. **d** Breakpoints identified at the replication barrier (750-bp product) using wild-type; a mixture of POLγ:A467T and POLγ:T251I+P587L; or T7 DNA polymerase. The H-strand sequence is shown and the various location of repeats at breakpoints are indicated in colour code (first repeat solid line, second repeat dashed). The frequency of each different breakpoint is indicated in the pie chart. Please note, the short repeat sequences next to the pie chart are shown as the L-strand sequences. **e** The majority (70%) of breakpoints identified for the 350-bp product using POLγ wild-type were localized to two imperfect repeats distant from the replication barrier. The frequency of sequence elements at breakpoints is indicated in the pie chart (for details, see Supplementary Table 2). Source data are provided as a Source Data file

subsequently unpairs from the stem-region and mispairs with shorter repeats in the 13447-CR region, downstream of the replication barrier. This observation explains why mutating 8470-CR has little effect on the formation of the 750-bp deleted product (Fig. 3c). We also analyzed deletion formation in vitro using equimolar concentrations of two mutated forms of POLγ, one with a single amino acid change (A467T) and one with two amino changes in the same allele (T251I + P587L). In vivo, these

mutated versions of POLγ, cause a compound heterozygous form of adPEO[23]. Interestingly, copy-choice recombination between the two 13-bp repeats were increased with the mutant polymerases (63% compared with 26% for POLγ wild-type) (Fig. 3d, Supplementary Table 2, Supplementary Fig. 6a, b, lanes 5 and 6). Deletions were also observed with T7 DNA polymerase (Supplementary Fig. 7a–d and Supplementary Table 2). No less than 83% of the deletions formed in the presence of T7 DNA

polymerase were due to perfect copy-choice recombination between the two 13-bp repeats (Fig. 3d). Our findings are in agreement with published data demonstrating that replicative polymerases may cause deletion by copy-choice recombination between direct repeats and that this effect is stimulated by secondary structures. As demonstrated here, this effect can be further accentuated by disease causing mutations in POLγ that impair processivity.

In our experiments, we repeatedly observed a 350-bp fragment (Figs 2d, f, h and 3c), which was independent of the 13-bp repeats (Fig. 3c, compare lanes 1–4). Sequencing revealed that the 350-bp product was due to copy-choice recombination between short repeats in the mitochondrial sequence that were distinct from the 13-bp common repeats (Supplementary Table 2 and Fig. 3e). Importantly, nearly all of the 5′ breakpoints identified in the 350-bp product had been previously observed in patients with mitochondrial deletion syndromes (Supplementary Table 2)[24–27]. All the breakpoints were enriched for nucleotide motifs that were rich in cytosines (e.g., CCC, ACCC, ACCCC) that were also frequently observed at breakpoints formed in vivo (compare Supplementary Table 2 and Fig. 3e with Fig. 1d). We observed one deletion formed between non-repeated regions (Supplementary Table 2, Class III). Interestingly, this specific class III breakpoint had previously been observed in vivo[6].

**POLγ-dependent deletion formation on a natural template.** We next investigated if deletions could be formed using templates lacking the artificial replication barrier (Fig. 4a, left panel). The single-stranded template used was the same as in previous experiments, but lacking the replication barrier hairpin sequence (Supplementary Table 1, wild-type template, T5). Following L-strand synthesis with POLγ in the presence of low mtSSB levels (Fig. 4b, lane 1), replication products were identified by PCR and

Southern blotting of the amplified region (Fig. 4c, lane 2). This analysis generated a band of the size expected for the common deletion (750 bp). The deleted product was lost when either 8470-CR or 13447-CR were mutated (Fig. 4a, b, T6 and T7), demonstrating that these two repeats are essential for deletion formation (Fig. 4c, compare lane 2 with 3 and 4).

To map the precise breakpoints, products from an L-strand synthesis reaction followed by a PCR were separated on agarose, and the barely detectable 750-bp product was isolated and sequenced (Fig. 4e, lane 2). For visualization, we also used products isolated from the first round of PCR as a template to perform a second round of PCR (Fig. 4e, lanes 3, 4). Nearly all sequenced clones (98%) represented a precise deletion between the two 13-bp repeats (Supplementary Table 2). One of the common repeats was retained, whereas the other was lost together with intervening sequences (Fig. 4f). Copy-choice recombination during L-strand synthesis in vitro could thus efficiently reconstitute the formation of the common deletion seen in mitochondrial diseases and in normal human ageing.

**Deletions formed in vitro are similar to those seen in vivo.** Finally, we analysed the products formed by copy-choice recombination in vitro by deep DNA sequencing, using the same bioinformatics pipeline used to characterize deletions in patients. This allowed us to look at a broader spectrum of deletions formed in vitro and not only the common deletion. We purified DNA from the copy-choice recombination assays (T1 and T5 templates) performed with wild-type or mutant POLγ (A467T/T251I + P587L) (Supplementary Fig. 6a, c). Our analysis identified a high number of mtDNA deletions (more than 1800 unique deletions) within the mtDNA sequence (Fig. 5a, Supplementary Data 2). We found deletions belonging to all three classes that had been identified in vivo, i.e., deletions between

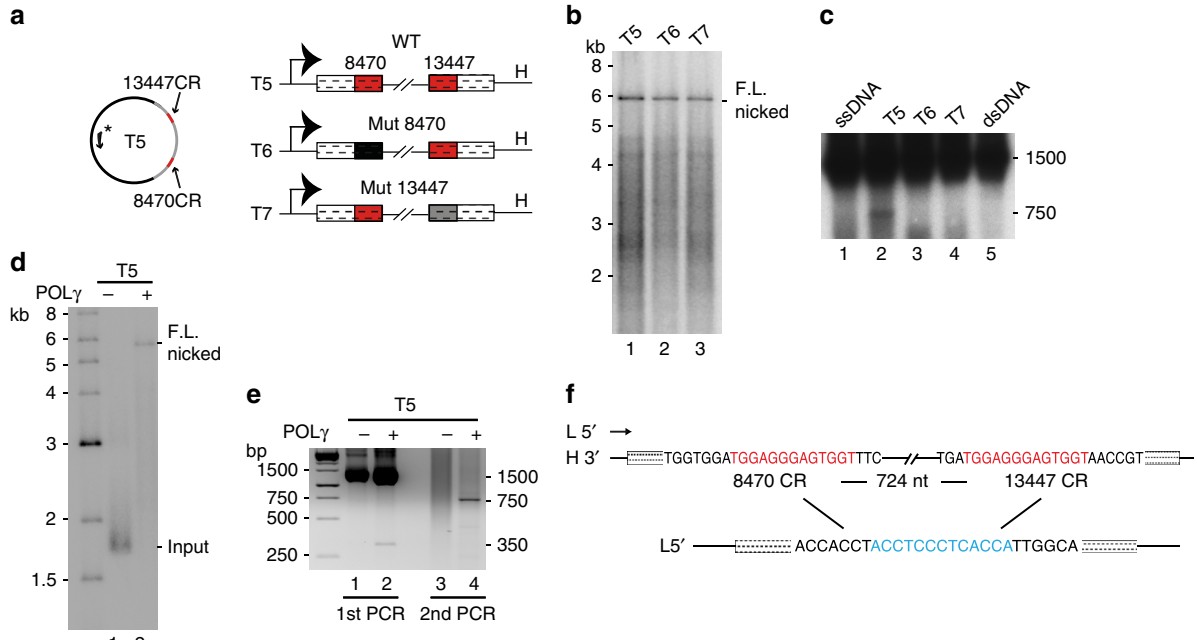

**Fig. 4** Common deletion formation during L-strand synthesis. **a** The wild-type template used is identical to that in Fig. 2a, but without the artificial replication barrier (T5). Mutant versions were created in which 8470-CR (T6) and 13447-CR (T7) had been mutated. For details see Supplementary Table 1. **b** L-strand DNA synthesis of the templates described in panel **a**. **c** The reactions in panel **b** were analysed by PCR followed by Southern blotting. The T5 template substrate was included in lane 1. Lane 5: double-stranded DNA control. **d** L-strand DNA synthesis as in panel **b** using the T5 template. **e** The reactions in panel **d** were analysed by PCR. After a first round of PCR (1st PCR) the products in the region around 750 bp were gel purified, followed by a second round of PCR (2nd PCR). **f** The vast majority (98%) of the 750-bp products corresponded to the common deletion. For details see Supplementary Table 2. Source data are provided as a Source Data file

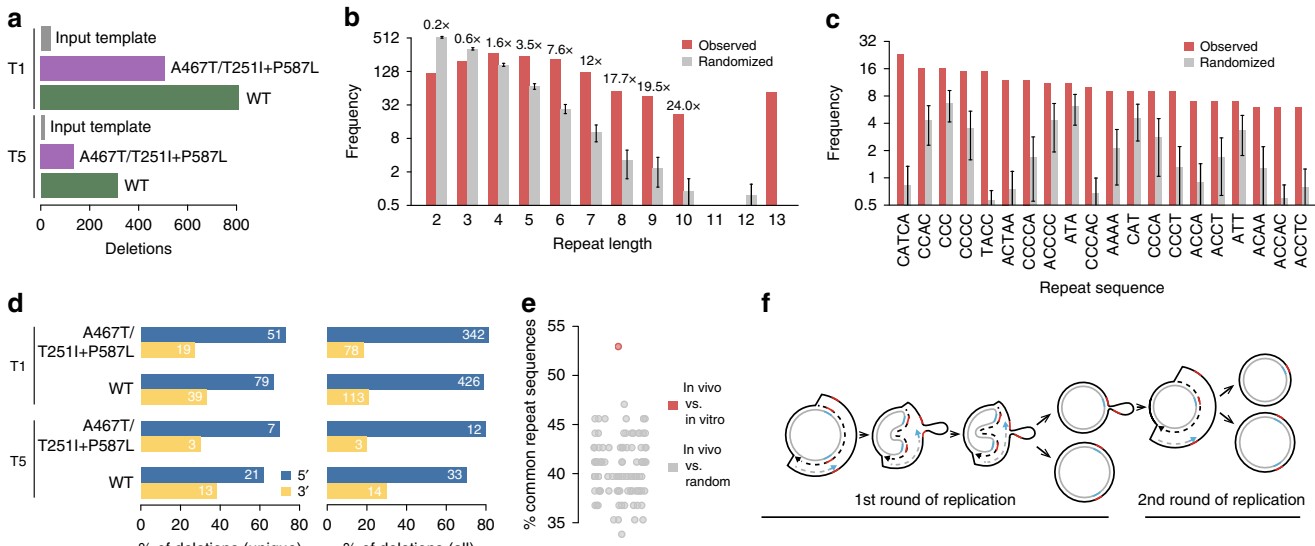

**Fig. 5** Similar mtDNA deletions formed in vitro and in vivo. **a** Frequency of deletions detected in mtDNA inserts (nt 8187–87784, 13,066–13,737) in T1 (barrier) and T5 (wild-type) templates replicated in vitro by indicated POLγ version. Non-replicated templates are included as controls (input template). **b** Frequencies of exact direct repeats overlapping or flanking each pair of breakpoints located within the mtDNA sequence in the templates, considering the longest match for each deletion (red bars). Results from randomized breakpoints are shown for comparison (grey bars), with error bars indicating the standard deviation (100 randomizations). **c** Frequencies for the most commonly observed repeat patterns in vitro using the breakpoints in T1 and T5 from reactions with POLγ wild-type or A467T/T251I+P587L. **d** Analysis of 5′ vs. 3′ retention of imperfect repeats, similar to Fig. 1e, for deletions formed in vitro. Results are shown for unique deletions as well as the complete set of deletions. **e** Non-random overlap between repeat sequences detected in vivo and in vitro. The fraction overlapping repeat patterns (minimum length 3 bp) in the in vitro and the in vivo data was determined (red), considering both templates and only breakpoints in the region common to both datasets. Results from 100 sets of randomly generated deletions are shown as controls (grey; P = 0.02, two-sided Wilcoxon test). **f** Schematic model of copy-choice recombination. The parental H-strand is represented in solid black with the repeats as red blocks. The parental L-strand is represented in solid grey with the repeats as blue blocks. The newly synthesized strands are in dashed lines. During SDM, replication repeats in the parental H-strand are exposed in their single-stranded conformation. Secondary structures in DNA may bring the repeats close to each other. If the replication machinery dissociates during replication of the first repeat, the 3′-end of the newly synthesised DNA can unpair from the first repeat and mispair with the second repeat. When DNA synthesis is restarted, a heteroduplex molecule is formed and the repeat closest to OriH will be lost together with intervening sequence. A deletion-containing double-stranded molecule is formed following a second round of mtDNA replication

direct repeats, imperfect repeats and non-repeat sequences. Exact sequence matches of length 5 bp or above were observed more frequently than expected and were enriched for nucleotide motifs that were rich in cytosines (Fig. 5b, c and Supplementary Data 2). A subset of the observed deletions (232) showed imperfect repeats that we analysed for directionality (Fig. 5d and Supplementary Data 2). Our analysis revealed that 158 and 74 of the informative cases (68.1% and 31.8%) supported retention of the repeat at the 5′ and 3′ side, respectively. These numbers changed to 813 and 208 (79.6% and 20.4%) when considering each supporting read individually rather than unique deletions. Taken together, the deletions formed in vitro were astonishingly similar to those observed in vivo (Fig. 1 and Supplementary Figs 3 and 4), and many of the repeat sequences (perfect direct repeats > 3 bases) were in fact shared (Fig. 5e). Our results strongly support copy-choice recombination during L-strand synthesis as a major cause for deletion formation in patients with mutations that affect mtDNA maintenance.

## Discussion

The model for mtDNA deletion formation presented here is a consequence of the distinct strand-displacement mode of mtDNA replication. In this model, L-strand DNA synthesis takes place on a long stretch of single-stranded H-strand DNA template (Fig. 1a), which may be particularly prone to copy-choice recombination. Following replication of a repeat sequence in the template H-strand, POLγ can stall and dissociate from the newly synthesized DNA-end. At this point, DNA breathing can

cause the 3′-end of the nascent L-strand to unpair from the DNA template, followed by reannealing and pairing with a second, downstream repeat sequence. When L-strand synthesis is resumed, one repeat and the intervening sequence will be lost (Fig. 5f and Supplementary Fig. 8 for details). It should be noted that this mechanism is strongly stimulated by repeat sequences, but can also apply to deletion formation between non-repeated sequences. Copy-choice recombination is also stimulated by conditions that promote stalling, e.g., mutations in proteins required for mtDNA replication or in those maintaining balanced nucleoside pools that affect processivity. Therefore, the model may explain why pathogenic variants in POLγ, TWINKLE, TOP3A and the nucleotide transporter, ANT1, can all stimulate mtDNA deletion formation[5]. However, if DNA synthesis becomes too inefficient, it will lead to mtDNA depletion rather than mtDNA deletions[28].

In our bioinformatics analysis of breakpoints formed in vivo, we identify a subset of imperfect repeats that are used to determine directionality. In the majority of cases, the 5′ repeat is retained, but there is also a significant portion for which the opposite is true. Even though we use stringent thresholds in our analysis, we can only predict the most likely breakpoints. The number of incorrect predictions is not known but could be significant. In support of this notion, please consider our in vitro reconstitution of deletion formation. In the reconstituted system, we only monitor L-strand synthesis. Therefore, copy-choice recombination can only, in theory, lead to the retention of the repeat close to OriL. However, bioinformatic analysis of deletion

products formed in vitro generates a directional bias similar to that observed in vivo, i.e., in about one third of the cases the 3′ repeat is predicted to be retained. We therefore believe that our bioinformatic analysis may underestimate the number of retained 5′ repeats and that the actual number can be higher than predicted. In other words, the existence of retained 3′-repeats at 25–35% of all breakpoints does not necessarily implicate the existence of alternative mechanisms for deletion formation. In addition, similar nucleotide motifs are enriched at breakpoints both in vivo and in vitro, adding further support for that the deletion formation process being the same in vivo and in vitro. In fact, many of the perfect direct repeats longer than three bases for which we cannot determine directionality, are shared between deletions formed in vivo and in vitro. It should be noted that our analysis does not allow us to distinguish the repeated generation of the same deletion, from clonal expansion of a single deleted parental molecule. However, we do not believe that this has any important bearing on our main conclusions, since the deletion patterns we describe have a high degree of diversity, in each case involving a spectrum of deletions having a large number of unique combinations of start/end positions.

Here, we have characterized how mtDNA deletions are formed in five different patients, with mutation in *POLG*, *TOP3A* or the *TWNK* genes. Even if these data support our proposed model, we cannot formally exclude that other mechanisms also exist. Future studies of additional patients will determine if also other mechanisms may contribute to deletion formation in specific cases. It is however worth noticing that our conclusions are in agreement with an earlier study that analysed a smaller number of pathological mtDNA deletions formed between imperfect repeats. The authors of this study also noted that 5′-repeats were preferentially retained[11]. This observation led them to question the validity of the earlier slipped-strand model for mtDNA deletion formation, which predicted that the repeat located close to OriH would be retained. Therefore, a second model was proposed, which states that deletions are formed during repair of double-stranded breaks in mtDNA[19]. This model has its own problems, since it depends on DNA repair and requires the existence of enzymatic activities that remain to be identified in the mammalian mitochondria. The DNA repair model also fails to explain why imperfect repeats located near OriL are preferentially retained.

The copy-choice recombination model suggested here provides a simple explanation for the many mtDNA deletions observed in vivo. Essential elements of the mechanism can be recapitulated in vitro with existing replication factors, and no additional enzymatic activities (i.e., repair enzymes) need to be invoked. The model can also explain sporadic mutation observed in healthy cells, even if this is an extremely rare event. In vivo, mtSSB binds and covers the displaced H-strand, preventing secondary-structure formation and thereby reducing the risk of deletion formation.

The copy-choice recombination model is also supported by a recent report demonstrating that deletion formation in vivo is dependent on active mtDNA replication[29]. Sfeir et al. induced breaks at specific loci in mtDNA using a number of different mito-TALEN constructs, and monitored the formation of the common deletion in vivo. As noted by the authors, the results were unexpected and, in part, inconsistent with previously published models for mtDNA deletion formation. However, their findings are in perfect agreement with the model proposed here. Formation of the common deletion was only triggered by a cut in the H-strand at the 3′-end of the 13-bp repeat close to OriL. In fact, this is the only mito-TALEN used in their study that would allow for copy-choice recombination between the two common repeats.

Finally, similar to previous reports[11,30], we noticed that the majority of OriH-proximal breakpoints are located near the end of the NCR. The molecular basis for this effect is unclear, but the 3′-end of the D-loop has been identified as a regulatory hub for mtDNA replication[31]. It is possible that secondary structures in this region make it more accessible for a free DNA 3′-end to anneal. Alternatively, structural elements in this region may stimulate recruitment of the replication machinery, thereby enhancing the chances for re-initiation of DNA synthesis. We will analyse these and other possibilities in future work.

## Methods

**Protein purification.** Recombinant baculoviruses encoding POLγB and the different POLγA versions were expressed in Sf9 cells[32]. These recombinant proteins all lacked the N-terminal mitochondrial targeting sequence and carried a carboxy-terminal 6 × His-tag. The proteins were purified over HIS-Select Nickel Affinity Gel (Sigma-Aldrich) and HiTrap Heparin HP (GE Healthcare), followed by HiTrap SP HP or HiTrap Q HP columns (GE Healthcare), depending on the net electrical charge of the protein. MtSSB lacking the N-terminal mitochondrial targeting sequence was expressed in insect cells and purified over DEAE Sepharose Fast Flow (GE Healthcare), HiTrap Heparin HP and HiTrap SP HP, followed by gel filtration using HiLoad Superdex 200 (GE Healthcare).

**Single-stranded DNA templates.** To create ssDNA templates for in vitro DNA replication, we used dsDNA templates derived from pBluescript SK and nicked the L-strand with the sequence-specific nicking enzyme Nb.BsmI (NEB). Following desalting by dialysis, the DNA containing the intact H-strand sequences was treated with exonuclease III (Thermo Fisher Scientific) and again dialyzed. The purity of the ssDNA was confirmed by electrophoresis on TAE agarose. A 5′ $^{32}$P-labelled oligonucleotide (5′-T$_{40}$ GGA TGA ACG AAA TAG ACA GAT CGC TGA GAT AG-3′) was annealed at a molar ratio of 2:1.

**L-strand DNA synthesis assay.** Unless otherwise indicated, reaction volumes were 20 μl, and reaction mixtures contained 5 nM template, 31 nM POLγA (wild-type or a 1:1 mixture of POLγ:A467T and POLγ:T251I + P587L), 37.5 nM POLγB (concentration calculated for a dimer) and 1.25 nM mtSSB (concentration calculated for a tetramer). The reactions also contained 20 mM Tris-HCl (pH 7.8), 1 mM DTT, 5 mM MgCl$_2$, 100 μg/mL bovine serum albumin (BSA), 10% glycerol, and 100 μM of each dNTP if nothing else was indicated. The reactions were incubated at 37 °C for 30–60 min, followed by heat inactivation for 10 min at 70 °C. Reactions were treated with 6 × DNA loading dye SDS solution (Thermo Fisher Scientific) before separation on a 0.8% TBE agarose gel containing 0.5 μg/mL EtBr and visualization by autoradiography. For T7 DNA polymerase (NEB), the standard reaction volume was 20 μl and the reaction mixture contained 5 nM template and 0.5 U T7 DNA polymerase, in 20 mM Tris-HCl (pH 7.5), 1 mM DTT, 10 mM MgCl$_2$, 100 μg/mL BSA, 10% glycerol and 250 μM of each dNTP. The reactions were treated and analyzed as described for POLγ. When indicated, the reactions were digested with High-Fidelity EcoRI (NEB) for 20 min at 37 °C before the analysis. All blots are also included as uncropped scans in data source file.

**PCR analysis and Southern blotting.** Standard PCR assays (20 μl) were performed with 2 U Taq DNA polymerase (NEB) in 1 × ThermoPol buffer, 200 μM dNTPs and 200 nM primers (M13 F: 5′-TGT AAA ACG ACG GCC AGT GAG-3′ and M13 R: 5′-TTC ACA CAG GAA ACA GCT ATG ACC-3′). The primer extension reaction was performed for 2 min at 95 °C, 20 s at 95 °C, 15 s at 56 °C, 90 s at 68 °C and 5 min at 68 °C with step 2–4 repeated in 35 cycles. The samples were separated on 1.5% agarose gel containing 0.5 μg/mL EtBr. Bands were either directly visualized under UV-light or gel purified, TA-cloned and analysed by DNA sequencing.

For Southern blotting analysis, standard PCR assays were performed as described above, blotted and hybridized with a PCR amplified probe against mtDNA.

**Analysis of mtDNA deletions using deep sequencing.** Total genomic DNA was isolated from muscle from three adult patients (PEO + phenotype) with compound heterozygous *POLG* variants using standard protocols. These are identified as follows: p.W748S/p.R1096C (patient 5 in ref. [33]); p.A467T/p.T251I+P587L (patient 10 in ref. [23]) p.A467T/p.X1240G+35aa (patient 3 in ref. [33]). Additionally, muscle from one adult patient with heterozygous dominant *TWNK* mutation (p.R357P), muscle from two healthy control patients and blood from two of the *POLG* patients (p.W748S/p.R1096C and p.A467T/p.T251I+P587L) were included as controls along with previously published data from a patient with DNA deletions caused by mutations in the *TOP3A* gene[17]. DNA was subjected to deep sequencing on a NextSeq 500 using the Nextera DNA library preparation kit (Illumina).

To identify mitochondrial deletions and duplications, alignment to chrM (rCRS assembly, NC_012920.1) and nuclear chromosomes (hg19 assembly) was initially performed using Bowtie2[34] (parameters: very-fast). Realignment of mitochondrial

and unaligned reads to chrM (using LAST[35] (lastdb parameters: uNEAR; lastal parameters: Q1–e80) allowed more sensitive identification of gapped alignments indicative of deleted or duplicated segments. LAST was chosen based on evaluations using simulated sequencing reads, where this tool was found to show higher accuracy in identification of breakpoints at single bp resolution compared with BLAT and BLAST. Post alignment, we filtered out likely PCR duplicates (mapped at identical positions when taking both mate reads into account, or at the same position in cases where the mate was filtered out in Bowtie2 alignment step). We also disregarded false gapped alignments that may arise in the D-loop region due to the circular mitochondrial genome being represented as a linear sequence. Additionally, gapped alignments having an E-value higher than 1e-5 or having more than one gap were filtered out. A minimum of 15 bp was required on both sides of the gap and a minimum gap length of 100 bp was required in the case of gapped alignments.

After realignment, we observed a chrM coverage depth of 2229×, 2218×, 1127× for the patients with compound heterozygous *POLG* variants (492721, 489871 and 249396 reads) along with 144× and 68× coverage for the blood controls (32709 and 16290 reads). The patient with *TOP3A* variants had a chrM coverage depth of 2608× (575731 reads), the patient with a pathogenic *TWNK* variant had a chrM coverage depth of 2974× (656190 reads), while the muscle controls were 846× and 1692×, respectively (187942 and 374999 reads).

The Bowtie2 alignment step was skipped for the in vitro samples (single-stranded DNA templates T1 and T5 of 4351 and 4295 bases) and the raw reads were aligned directly to the respective template sequences using LAST followed by identification of deleted or duplicated regions as described above. Three samples were analyzed for each template, them being (1) non-replicated (2) template replicated with wild-type POLγ and (3) template replicated with mutant POLγ (A467T/T251I+P587L). A coverage depth of 775486×, 660922× and 690030× was observed for T1 template (44827147, 38251558 and 39916049 reads) while the samples for T5 template showed 542977×, 734095× and 721449× coverage (30992644, 41942787 and 41219507 reads).

Gapped alignments, indicative of deletions/duplications, were next clustered such that segments having identical breakpoints on both ends were grouped together. For visualization (Fig. 1b), more flexibility was allowed: segments were clustered using single-linkage clustering using a distance threshold of 50 bp, considering the maximum distance exhibited by the two breakpoints, and only clusters containing at least two reads were considered. It should be noted that there is an inherent ambiguity when it comes to deletions and duplications, since deletion of one specific mtDNA arc and duplication of the complementary arc will give rise to the same type of chimeric reads and hence similar gapped alignments, but one or the other classification is normally unlikely due to disruption of one or both origins of replication (Supplementary Fig. 9). While initially classified as deletions, segment clusters were therefore reclassified as likely duplications in case of overlap with replication origins. Heteroplasmy was determined by calculating the fraction of gapped to wild-type reads at each breakpoint, and by considering the average values for the two breakpoints for a given segment cluster.

Direct repeats were analyzed by determining the longest exact sequence match overlapping with the two breakpoints for a given deletion. To determine whether repeats were typically retained on the 5′ or 3′ side of deletions, we focused on a subset of cases with sufficient differences in the repeat sequence at the two ends such that the position of the breakpoints could be safely deduced by the aligner, specifically allowing a maximum uncertainty of 1 bp. We then determined the size of the longest repeat on either side while allowing one mismatch. Based on this, deletions were classified as having either 5′ or 3′ retained repeats, while requiring a difference in repeat length of > = 2 to avoid forcing a classification in cases where both ends scored too similarly.

**Ethical approval.** Informed consent for diagnostic and research studies was obtained for all subjects in accordance with the Declaration of Helsinki protocols and approved by the local institutional review board (Newcastle and North Tyneside Local Research Ethics Committees (reference 2002/205)).

**Reporting summary.** Further information on experimental design is available in the Nature Research Reporting Summary linked to this article.

## Data availability

Sequencing data from patient samples are available through the European Genome-phenome Archive (EGA), accession number EGAS00001003148, while sequencing ?data from in vitro experiments are available through the European Nucleotide Archive (ENA), accession number PRJEB27814. The code used for identifying deletions and duplications from sequencing data is available at: http://bit.ly/mipt_v1. Source data for Figs 2c–h, 3b, c, 4b–e, as well as for Supplementary Figures 5a, 6a–c and 7a–d are provided as a Source Data file. A reporting summary for this Article is available as a Supplementary Information file.

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

## Acknowledgements

We thank Dr. Enrique Viguera for useful discussions about the DNA construct used in this study and Dr Jay P. Uhler who prepared the illustration in Fig. 5. This work was supported by the Swedish Research Council (M.F., C.M.G., E.L.); the Swedish Cancer Foundation (M.F., C.M.G., E.L.); the European Research Council (M.F.); the IngaBritt and Arne Lundberg Foundation (M.F., C.M.G.); the Knut and Alice Wallenberg Foundation (M.F., C.M.G., E.L.); grants from the Swedish state under the agreement between the Swedish government and the county councils, the ALF agreement to CMG (ALFGBG-728151) and MF (ALFGBG-727491); the Swedish Foundation for Strategic Research (E.L.); the Wenner-Gren Foundation (E.L.); the Wellcome Centre for Mitochondrial Research (203105/Z/16/Z), the Medical Research Council (MRC) Centre for Translational Research in Neuromuscular Disease, Mitochondrial Disease Patient Cohort (UK) (G0800674), the UK NIHR Biomedical Research Centre for Ageing and Age-related disease award to the Newcastle upon Tyne Foundation Hospitals NHS Trust, the MRC/EPSRC Molecular Pathology Node and the UK NHS Highly Specialised Service for Rare Mitochondrial Disorders of Adults and Children (R.M., R.W.T.). The computations were in part performed on resources provided by SNIC through Uppsala Multi-disciplinary Center for Advanced Computational Science (UPPMAX) under Project SNIC 2017/11-3.

## Author contributions

Designed research: Ö.P., Y.M., S.B., E.L., C.M.G. and M.F.; performed research: Ö.P., Y.M., S.B. and L.J.; contributed analytical tools: Ö.P., Y.M., A.K.B., S.B., L.J., R.M. and R.W.T.; analysed data: Ö.P., Y.M., L.J., J.U., S.B., E.L. and M.F.; wrote the paper: Ö.P., Y.M., J.U., S.B., E.L., C.M.G. and M.F.

## Additional information

**Competing interests:** The authors declare no competing interests.

