## [Peer Review File · Nature Communications]

Reviewers' comments:

Reviewer #1 (Remarks to the Author):

Copy-choice recombination during mitochondrial L-strand synthesis causes DNA deletions: Persson et al.

In this paper the authors present an interesting model to explain why most deletions in mitochondrial DNA occur within the so-called 'major-arc'. The paper presents very elegant in vitro assays to show that copy-choice recombination could be one mode by which deletions are formed. Although this is an elegant and interesting model, the paper contains several mistakes, omissions and possibly biased interpretations also of published data and leaves me with quite a few questions.

Comments:

1-My first concern is the analysis of repeats (Suppl Table I). In my opinion this analysis seems to be done somewhat carelessly. For many of the deletions that are shown to have directionality (by blue colour coding) the deletion window can be shifted, often resulting in a shift in directionality, i.e. the repeat close to OH is retained and not the one close to OL. (A PDF file with these comments will be attached to illustrate the point using the same colour coding as the authors have used).

To give some examples from Suppl Table I

```
10938 14529 3591 AACCTTTTCCTCCGACCC[CCTAACAACCCCC  
TATTAAACCCATATAACC]TCCCCCAAATTC
```

(This line is btw duplicated in the Table)

Alternate interpretation

```
10938 14529 3591 AACCTTTTCCTCCGAC[CCCCTAACAACCCCC  
TATTAAACCCATATAA]CCTCCCCCAAATTC
```

```
7361 13235 5874 AAAGTCCTAATAGTAGAA[GAACCCTCCATAA  
ACAAAATGACATCAAAAA]AATCGTAGCCTTC
```

Alternate interpretation

```
7361 13235 5874 AAAGTCCTAATAGTAG[AAGAACCCTCCATAA  
ACAAAATGACATCAAA]AAAATCGTAGCCTTC
```

The point here is that the rules that apply to the OL side should also apply to the other side and it should be done consistently for all sequences. One could also ask whether the sequence directly at the breakpoint is what is important if close to the breakpoint (give or take 20 nt or even more) a longer but perhaps less perfect repeat is present. Also if one looks carefully, longer (imperfect) repeats might also be present by shifting the deletion window, such as the example below. It is not clear on what basis this should be rejected if a 4 bp imperfect repeat is allowed (such as cccc versus cacc). Again, clear rules should be defined and consistently applied to the analysis. Finally, based on the analysis presented in Fig 1c-d, one could question if 3 nucleotide repeats should be

included in the analysis at all. Overall, the repeat data analysis is not so unambiguous as the authors lead us to believe.

```
10940 16072 5132 CCTTTTCCTCCGACCCCC[TAACAACCCCCCT  
CCCAAGTATTGACTCACC]CATCAACAACCGC
```

Alternate interpretation

```
10940 16072 5132 CCTTTTCCTCCGACCC[CCTAACAACCCCCCT  
CCCAAGTATTGACTCA]CCCATCAACAACCGC CC(T)A(A)CAAC
```

2-Given the proposed model one would expect to frequently find the common deletion in patient samples or find breakpoints centered around the 13-bp common deletion repeats? This data can not be extracted from this paper. One might also expect, based on the proposed model and on the severity of the POLG mutations, that the breakpoint distribution differs between patient samples, being either closer or further away from OL, dependent on the propensity of POLG to stall. Although Fig 1 might give an indication, I would like to see that the authors present histograms akin to the analysis done in Ref 10, as it would be much more informative. In addition, the authors should include the entire deletion dataset for each patient, including the exact sequences and the number of these sequences detected for each unique deletion, so that other researchers can independently mine/analyze these data. This can of course be a supplementary file (e.g. in Excell format or a more extended version of Suppl table 1).

3- My first remark in point 2 also relates to the fact that in the common deletion 1989 paper, which the authors cite, it is suggested that in that case the repeat close to OriH is retained. This is of course somewhat awkward given that the in vitro assays in this submission are based on this common deletion and their flanking repeats. It illustrates a weakness in the current in vitro approach which is that it is not a fully reconstituted system but rather uses a single stranded template, with only lagging strand replication initiating from OriL. The approach is therefore biased. Ideally the authors should use a reconstituted system that starts with dsDNA and includes also the TWINKLE helicase and allows for both leading and lagging strand synthesis. This of course will be a very daunting task and I wouldn't expect that for this paper, but the bias should be discussed, as well as the fact that the repeat close to OriH has been proposed to be retained previously. Perhaps also an analysis of all common deletions published points to something different, since the 1989 paper of course only had 1 sample.

Based on the fact that the mtDNA replication system is not fully reconstituted in vitro and on the basis of my other comments, I find the statement at the end of the discussion far too strong.

'Our data argues against the previously proposed models and instead demonstrate that mitochondrial deletions are formed by copy choice recombination during active L-strand DNA synthesis.'

'suggest' would be a more appropriate choice of words

4-As observed by many others that have analyzed multiple deletions from patient samples, also in this submission there is a strong preponderance for the OH-proximal deletion breakpoint at the end of the NCR, around bp 16070, this in contrast to single sporadic deletions that have been reported. It is unclear how the proposed model can explain this observation. I find it strange that the authors have not considered this particular aspect of their sequence analysis, but perhaps in a resubmission they can speculate and/or explain on the basis of their model.

5-On page 9 the authors state the following:

'Our conclusions are also in perfect agreement with an earlier study that analysed a smaller number of pathological mtDNA deletions formed between imperfect repeats. In this study, the repeat close to OriH was removed in 94% of the cases in which directionality could be determined, which is similar to the number we found in our patients 10.'

Although ref 10 indeed states that the 5' (OL) repeat is preferentially retained (see quote below), the statement is not at all so unambiguous that it is allowed to state that in 94% of cases the 3' (OH) repeat was deleted.

From: Samuels, Schon and Chinnery (ref 10).

'The analysis of deletions formed between imperfect direct repeats (n=63) showed that in 44% of cases the 3' direct repeat was deleted, in 52% of cases the deletion breakpoint was within the direct repeat, retaining both 5' and 3' components but the 5' repeat was only removed completely in 3% of cases. Although the deletion of both elements has been noted before [4,37], our analysis shows that the 5' repeat is retained preferentially in the deleted molecule. This bias would not be expected if the deletions were formed by simple intramolecular recombination involving the association of repeat sequences.'

Other comments:

Pg 3: 'paternal H-strand' should be 'parental H-strand'

Suppl Figure 3: the three nucleotides 3' of the 13447 repeat (TGA) are incorrect, it should read AGT.

Reviewer #2 (Remarks to the Author):

The key observation is based on deep sequencing three muscle samples and a comparison to one control cell line.

The bioinformatic methods are superficial, which makes the evaluation of the data impossible. I find it extremely difficult to believe that all of the reads mapped beautifully to the major arc shown in Fig1. Gapped alignment approaches are not 'perfect', and there will be some misaligned reads etc. creating 'noise' in the analysis. Where is this noise? And how did the authors deal with this.

A major concern – particularly given the new proposed model of deletion formation – is that the apparent breakpoint sequence reads are generated as an artefact of the sequencing method and/or the alignment process. PCR amplification of the tagged reads could generate recombinants in vitro. The single experiment carried out on the human cell line is reassuring, but a better control would be blood DNA from the same patient which was extracted using the same method at the same time. It is known that extraction procedures and/or storage can lead to DNA damage and the potential for artefactual deletions.

A good control would be to deep sequence the templated they generated for their in vitro work – essentially to test whether the sequences known to facilitate deletions lead to artefactual recombinants simulating deletions in vitro.

The authors then comment on the relative abundance of different types of deletions – but it is not clear to me that duplicate reads were effectively excluded, or whether there was the potential for biased application of the apparent fusion.

Clonal expansion is a further complication. It is possible that – for example – the slippage mechanism is preferable (but not exclusive) for deletion formation, but that clonal expansion preferentially affects deletions that look as though they were generated by copy choice recombination. Only the clonally amplified deletions will be sequenced. The authors could get around this by using bar-code single molecular tagging and extreme high depth.

Finally, there is one large caveat here – that the deletions were detected in patients with a mutated DNA polymerase. It is possible that the pattern of deletions would be different in other contexts.

The opening paragraph of the discussion recapitulates the introduction.

Reviewer #3 (Remarks to the Author):

This paper investigates the occurrence and cause of mtDNA deletions. First, mtDNA deletions were identified from muscle samples of three patients with POLG mutations and a POLG disease. Secondly, to recapitulate these deletions, the authors design a substrate that mimics replication of the L-strand synthesis. After replication by wildtype POLG and PCR amplification the authors can identify specific deletion that occur between the direct repeats and potentiated by secondary structures. The deletions identified fit the experimental design, suggesting a similar mechanism described for *E. coli*, copy-choice recombination.

Major comments

There are three major issues with this paper

1. I doubt there is anything special about POLG as compared to other replicative DNA polymerases that dictate deletion formation and it is likely any polymerase will cause a deletion at direct repeats and secondary structure. It's more about the DNA structure than the polymerase.
2. There is a disconnect between the in vivo analysis and the in vitro analysis. The in vivo analysis identifies mtDNA deletions in patients with POLG mutations, but no healthy or wt controls are investigated. But the in vitro analysis attempts to recapitulate these deletions with only the WT POLG. Can the authors identify deletions in age matched healthy controls, and can the authors demonstrate enhanced deletion formation in vitro with a recombinant POLG containing these mutations?
3. It's unclear about certain controls, such as PCR amplification of the T1 substrates without POLG and using T7 Pol with thioredoxin.

Specific comments:

A normal age matched control for the in vivo deletion analysis was not performed. This analysis needs a healthy control.

I don't think there is anything special about POLG that contributes to the formation of deletions, which suggests it's more about the DNA content and structure that dictates these deletions. To help support that, or not, the authors should investigate other Family A polymerases. A T7 control was included in the supplement, but not compared side by side with POLG throughout many of these experiments.

Fig 2d, There needs to be a POLG minus control in this experiment to account for deletions generated by the Taq PCR application, especially since Taq is well known to make similar errors in vitro.

The in vitro analysis focuses on POLG disease patients while the in vitro uses only the wild type

POLG. Can the authors also test a recombinant POLG in vitro with any of these disease mutations to demonstrate enhanced deletion formation.

Could the authors clarify what is meant by "low" mtDNA SSB levels used in Fig 4B. What is the molecular stoichiometry of the SSB with DNA.

Fig 4b doesn't show any difference between these DNA substrates.

It appears that Supplementary fig 2 with T7 DNA Pol, in which the authors state has a weaker strand-displacing activity, was done in the absence of thioredoxin. If the authors are truly comparing strand displacement activities related to deletion formation, then they should include thioredoxin, similar to POLG2 as a processivity factor.

REVIEWER #1.

#1. The reviewer writes:

My first concern is the analysis of repeats (Suppl Table I). In my opinion this analysis seems to be done somewhat carelessly. For many of the deletions that are shown to have directionality (by blue colour coding) the deletion window can be shifted, often resulting in a shift in directionality, i.e. the repeat close to OH is retained and not the one close to OL. (A PDF file with these comments will be attached to illustrate the point using the same colour coding as the authors have used).

To give some examples from Suppl Table I

10938 14529 3591

AACCTTTTCCTCCGACCC [CCTAACAAACCCCCTATTAAACCCATATAACC] TCCCCCAAATTC

(This line is btw duplicated in the Table)

Alternate interpretation

10938 14529 3591

AACCTTTTCCTCCGAC [CCCCTAACAAACCCCCTATTAAACCCATATAA] CCTCCCCCAAATTC

7361 13235 5874

AAAGTCCTAATAGTAGAA [GAACCCTCCATAAACAAAATGACATCAAAAA] AATCGTAGCCTTC

Alternate interpretation

7361 13235 5874

AAAGTCCTAATAGTAG [AAGAACCCTCCATAAACAAAATGACATCAAA] AAAATCGTAGCCTTC

The point here is that the rules that apply to the OL side should also apply to the other side and it should be done consistently for all sequences. One could also ask whether the sequence directly at the breakpoint is what is important if close to the breakpoint (give or take 20 nt or even more) a longer but perhaps less perfect repeat is present. Also if one looks carefully, longer (imperfect) repeats might also be present by shifting the deletion window, such as the example below. It is not clear on what basis this should be rejected if a 4 bp imperfect repeat is allowed (such as cccc versus cacc). Again, clear rules should be defined and consistently applied to the analysis. Finally, based on the analysis presented in Fig 1c-d, one could question if 3 nucleotide repeats should be included in the analysis at all. Overall, the repeat data analysis is not so unambiguous as the authors lead us to believe.

10940 16072 5132

CCTTTTCCTCCGACCCCC [TAACAACCCCCCTCCCAAGTATTGACTCACC] CATCAACAACCGC

Alternate interpretation

10940 16072 5132

CCTTTTCCTCCGACCC [CCTAACAAACCCCCTCCCAAGTATTGACTCA] CCCATCAACAACCGC

CC (T) A (A) CAAC

Our response:

We can only agree with this criticism, and we are thankful to the reviewer for pointing this out. To avoid any biases and errors, we have avoided manual curation in the new version of the manuscript. Instead, we have set up an automated process to determine which end is likely to

have been retained in a given deletion, in the subset of cases where this can be deduced from sequence.

In brief, we considered only cases with sufficient differences in the repeat sequence at the two ends such that the position of the breakpoints could be safely deduced (1 bp flexibility allowed). For example, below is an excluded case where the aligner has no way of knowing what end is being retained or deleted:

```
Initial alignment:
CAAATGACATCAAAA [AAATCGTAGCCTTCT...AATACTAAACCCCAT] AAATAGGAGAAGGCT
3' repeat retained:
CAAATGACATCAAAA [AAATCGTAGCCTTCT...AATACTAAACCCCAT] AAATAGGAGAAGGCT
5' repeat retained:
CAAATGACATCAAAAAAAT [CGTAGCCTTCT...AATACTAAACCCCATATAAT] AGGAGAAGGCT
```

In contrast, in the case below, the genomic positions of the breakpoints are unambiguous to within 1 bp:

```
Initial alignment:
CCACCAACCCCCCT [CCCCGCTTCTGGCC      CCCAAAGACACCCCC] CACAGTTTATGTAGC
5' repeat retained
CCACCAACCCCCCTC [CCCCGCTTCTGGCC      CCCAAAGACACCCCC] ACAGTTTATGTAGC
```

In cases like the latter, we determined the size of the longest repeat on either side while allowing one bp mismatch. Based on this, deletions were classified as having either 5' or 3' retained repeats. A difference in repeat length of ≥ 2 was required, to avoid forcing a classification in cases where both ends scored too similarly.

Of all unique deletions ($n = 623$), we were able to assign a class for 20%. Of these, 67% and 33% were predicted to have 5' and 3' repeats retained, respectively. In fact, this may be an underestimation since recurring deletions at hot spots are calculated only once. If we instead consider each deletion-supporting read as an independent event ($n = 780$), these numbers improved to 75/25%. Analysis of randomly generated breakpoints (100 simulations of similar size as the observed data) yielded ratios that were 50/50 on average and never as skewed as the observed data, showing that the method has no inherent 5' or 3' bias.

Thus, using an unbiased computational classification, the deletions observed in vivo show a clear preference for retention of repeats at the 5' side (OriL-proximal position). These results have been included in Fig. 1E, and the Result section has been updated accordingly (see page 4).

#2. The reviewer writes:

Given the proposed model one would expect to frequently find the common deletion in patient samples or find breakpoints centered around the 13-bp common deletion repeats? This data cannot be extracted from this paper. One might also expect, based on the proposed model and on the severity of the POLG mutations, that the breakpoint distribution differs between patient samples, being either closer or further away from OL, dependent on the propensity of POLG to stall. Although Fig 1 might give an indication, I would like to see that the authors present histograms akin to the analysis done in Ref 10, as it would be much more informative. In addition, the authors should include the entire deletion dataset for each patient, including the exact sequences and the number of these sequences detected for each unique deletion, so that other researchers can independently mine/analyze these data. This can of course be a supplementary file (e.g. in Excell format or a more extended version of Suppl table 1).

Our response:

We thank the reviewer for these useful suggestions. We have plotted the histograms of breakpoint positions and included these as Supplementary Fig. 1. A detailed supplementary table have been added, which lists the identified breakpoints (Supplementary Table 1).

On page 4, first paragraph, we now write:

“The vast majority of deletions detected (total 623) were in the major arc, located in the region between OriH and OriL (Fig. 1a-b and shown as histograms of breakpoint positions in Supplementary Fig. 1). Similar to previous reports, we noticed that the majority of OriH-proximal breakpoints clustered near the end of the non-coding region. All identified and analysed deletions can be found in Supplementary Table 1.”

#3. The reviewer writes:

My first remark in point 2 also relates to the fact that in the common deletion 1989 paper, which the authors cite, it is suggested that in that case the repeat close to OriH is retained. This is of course somewhat awkward given that the *in vitro* assays in this submission are based on this common deletion and their flanking repeats. It illustrates a weakness in the current *in vitro* approach which is that it is not a fully reconstituted system but rather uses a single stranded template, with only lagging strand replication initiating from OriL. The approach is therefore biased. Ideally the authors should use a reconstituted system that starts with dsDNA and includes also the TWINKLE helicase and allows for both leading and lagging strand synthesis. This of course will be a very daunting task and I wouldn't expect that for this paper, but the bias should be discussed, as well as the fact that the repeat close to OriH has been proposed to be retained previously. Perhaps also an analysis of all common deletions published points to something different, since the 1989 paper of course only had 1 sample.

Based on the fact that the mtDNA replication system is not fully reconstituted *in vitro* and on the basis of my other comments, I find the statement at the end of the discussion far too strong.

‘Our data argues against the previously proposed models and instead demonstrate that mitochondrial deletions are formed by copy choice recombination during active L-strand DNA synthesis.’

‘suggest’ would be a more appropriate choice of words

Our response:

We followed the reviewer's suggestion and changed “demonstrate” to “suggest” (see page 3. last paragraph).

About the shortcomings of our analysis. We are actively working with developing a complete system with simultaneous H- and L-strand DNA synthesis to study deletion formation. There are still a number of hurdles to be overcome before this is possible. There are however certain benefits of only looking at L-strand synthesis in a reconstituted system. As requested by the reviewer, we now discuss this point and the shortcomings of our analysis in the discussion section of the paper. We also discuss the implications of our *in vitro* findings and the similarities with the breakpoints observed *in vivo*.

On page 10, second paragraph, we now write.

“In our bioinformatics analysis of breakpoints formed *in vivo*, we identify a subset of imperfect repeats which are used to determine directionality. In the majority of cases, the 5' repeat is

retained, but there is also a significant portion for which the opposite is true. Even though we use stringent thresholds in our analysis, we can only predict the most likely breakpoints. The number of incorrect predictions is not known but could be significant. In support of this notion, please consider our *in vitro* reconstitution of deletion formation. In the reconstituted system, we only monitor L-strand synthesis. Therefore, copy-choice recombination can only, in theory, lead to the retention of the repeat close to OriL. However, bioinformatic analysis of deletion products formed *in vitro* generates a directional bias similar to that observed *in vivo*, i.e. in about one third of the cases the 3' repeat is predicted to be retained. We therefore believe that our bioinformatic analysis may underestimate the number of retained 5' repeats and that the actual number may be higher than predicted. In other words, the existence of retained 3'-repeats at 25 – 35% of all breakpoints do not necessarily implicate the existence of alternative mechanisms for deletion formation. In addition, similar nucleotide motifs are enriched at breakpoints both *in vivo* and *in vitro*, adding further support for that the deletion formation process being the same *in vivo* and *in vitro*. In fact, many of the perfect direct repeats >3 bases for which we cannot determine directionality, are shared between deletions formed *in vivo* and *in vitro*.”

#4. The reviewer writes:

As observed by many others that have analyzed multiple deletions from patient samples, also in this submission there is a strong preponderance for the OH-proximal deletion breakpoint at the end of the NCR, around bp 16070, this in contrast to single sporadic deletions that have been reported. It is unclear how the proposed model can explain this observation. I find it strange that the authors have not considered this particular aspect of their sequence analysis, but perhaps in a resubmission they can speculate and/or explain on the basis of their model.

Our response:

As pointed out, we did not speculate and/or explained why there is a strong preponderance for OriH-proximal deletion breakpoints at the end of the NCR. Presently, we do not have an easy explanation for this effect, but in the new version of the manuscript, we discuss some possibilities.

On page 12, last paragraph, we now write

“Finally, similar to previous reports ^{11,29}, we noticed that the majority of OriH-proximal breakpoints are located near the end of the NCR. The molecular basis for this effect is unclear, but the 3'-end of the D-loop has been identified as a regulatory hub for mtDNA replication ³⁰. It is possible that secondary structures in this region make it more accessible for a free DNA 3'-end to anneal. Alternatively, structural elements in this region may stimulate recruitment of the replication machinery, thereby enhancing the chances for re-initiation of DNA synthesis. We will analyse these and other possibilities in future work.”

#5. The reviewer writes

On page 9 the authors state the following:

‘Our conclusions are also in perfect agreement with an earlier study that analysed a smaller number of pathological mtDNA deletions formed between imperfect repeats. In this study, the repeat close to OriH was removed in 94% of the cases in which directionality could be determined, which is similar to the number we found in our patients 10.’

Although ref 10 indeed states that the 5' (OL) repeat is preferentially retained (see quote below), the statement is not at all so unambiguous that it is allowed to state that in 94% of cases the 3' (OH) repeat was deleted.

From: Samuels, Schon and Chinnery (ref 10).

'The analysis of deletions formed between imperfect direct repeats (n=63) showed that in 44% of cases the 3' direct repeat was deleted, in 52% of cases the deletion breakpoint was within the direct repeat, retaining both 5' and 3' components but the 5' repeat was only removed completely in 3% of cases. Although the deletion of both elements has been noted before [4,37], our analysis shows that the 5' repeat is retained preferentially in the deleted molecule. This bias would not be expected if the deletions were formed by simple intramolecular recombination involving the association of repeat sequences.'

Our response:

We have clearly made a mistake here. We have completely rephrased this section and on page 11, second paragraph, we now write:

"Our conclusions are in agreement with an earlier study that analysed a smaller number of pathological mtDNA deletions formed between imperfect repeats. The authors of this study also noted that 5'-repeats were preferentially retained¹⁰. This observation led them to question the validity of the earlier slipped-strand model for mtDNA deletion formation which predicted that the repeat located close to OriH would be retained."

#6. The reviewer writes

Pg 3: 'paternal H-strand' should be 'parental H-strand'

Our response:

Corrected!

#7. The reviewer writes

Suppl Figure 3: the three nucleotides 3' of the 13447 repeat (TGA) are incorrect, it should read AGT.

Our response:

Corrected!

REVIEWER #2.

#1. The reviewer writes:

The key observation is based on deep sequencing three muscle samples and a comparison to one control cell line.

The bioinformatic methods are superficial, which makes the evaluation of the data impossible. I find it extremely difficult to believe that all of the reads mapped beautifully to the major arc shown in Fig1. Gapped alignment approaches are not 'perfect', and there will be some misaligned reads etc. creating 'noise' in the analysis. Where is this noise? And how did the authors deal with this.

Our response:

We agree that the methods description was too sparse and have now considerably expanded this section. The problem is indeed challenging, and we have put significant effort into establishing and optimizing this pipeline. This ensured that false positive breakpoints were minimized, and in the end we believe that our method performs very well. A discussion about the technical aspects of the pipeline is included further below.

Importantly, two new negative control muscle samples have been included in the revised manuscript, presented in Fig. 1B. While comparable in terms of coverage to the *POLG* muscle samples (1500-2000X), very few deletions/duplications were detected in these samples. Additionally, zero breakpoints were found in two blood controls with lower coverage (from patients POLG 748/1096 and POLG 467/251+587), also included in the revised manuscript and now mentioned in Results. Crucially, the muscle and blood control samples serve the dual purpose of 1) controlling for false positives that may arise due to technical issues, such as bioinformatics-related aspects (e.g. alignment) or sequencing artefacts, and 2) controlling for the general biological properties of muscle mtDNA, thereby ensuring that the deletions seen in *POLG* mutant samples truly relate to *POLG*. These results thus strongly argue against false positives being a major issue in our analysis.

As described in the clarified Methods description, our pipeline involves (1) alignment of reads to the nuclear + mitochondrial genomes using a fast aligner (bowtie2), (2) extraction of unmapped reads and reads mapped to the mitochondrial genome, and (3) precision realignment of the extracted reads using a slow aligner (LAST), using a stringent E-value threshold ($1e^{-5}$) and crucially requiring at least 15 bp alignment length on each side in a gapped alignment. We spent considerable effort testing different tools and parameters while evaluating the results on simulated reads from mitochondrial genomes with *in silico* generated deletions. The simulations were crucial and allowed us to eventually arrive at a workflow that performed well in terms of both sensitivity and specificity.

#2. The reviewer writes:

A major concern – particularly given the new proposed model of deletion formation – is that the apparent breakpoint sequence reads are generated as an artefact of the sequencing method and/or the alignment process. PCR amplification of the tagmented reads could generate recombinants *in vitro*. The single experiment carried out on the human cell line is reassuring, but a better control would be blood DNA from the same patient which was extracted using the same method at the same time. It is known that extraction procedures and/or storage can lead to DNA damage and the potential for artefactual deletions.

Our response:

We would argue that the best control would be the same kind of tissue (muscle), sequenced at comparable depth, but from individuals who are wild type with respect to *POLG* and do not suffer from a mtDNA deletion syndrome. As described above (comment 1), we have now sequenced two such samples while finding very few deletions, thus efficiently arguing against experimental or alignment artefacts being a major concern. These data are included in the new version of the manuscript.

We have also sequenced blood DNA from two of the patients (POLG W748S/R1096C and POLG A467T/T251I+P587L). No breakpoints were detected in these samples. As expected, mtDNA coverage was lower in these samples compared to muscle DNA.

#3. The reviewer writes:

A good control would be to deep sequence the templated they generated for their in vitro work – essentially to test whether the sequences known to facilitate deletions lead to artefactual recombinants simulating deletions in vitro.

Our response:

We have followed this suggestion and sequenced the template. In response to the concerns raised by reviewer 3, we have also performed next generation sequencing of the replication products formed by wild type and muted POL γ (A467T/T251I+P587L) in vitro. The coverage was high (~7800 \times) and the generated data were analysed using the same pipeline as for the mtDNA derived from muscle tissue. The template samples had very low abundance of breakpoints compared to the replicated plasmid DNA samples. These results have been included in the manuscript as Fig. 5A.

#4. The reviewer writes:

The authors then comment on the relative abundance of different types of deletions – but it is not clear to me that duplicate reads were effectively excluded, or whether there was the potential for biased application of the apparent fusion.

Our response:

While estimating the frequency of repeats we have been careful to remove duplicated reads (gapped alignments with the same fragment size and position) to only consider a unique set of reads for each sample. Furthermore, our pipeline effectively removes duplicated reads before clustering reads to estimate heteroplasmy for a given deletion.

#5. The reviewer writes:

Clonal expansion is a further complication. It is possible that – for example – the slippage mechanism is preferable (but not exclusive) for deletion formation, but that clonal expansion preferentially affects deletions that look as though they were generated by copy choice recombination. Only the clonally amplified deletions will be sequenced. The authors could get around this by using bar-code single molecular tagging and extreme high depth.

Our response:

Our study demonstrates that a simple, well-established model for deletion formation shown in multiple other systems can explain how mtDNA deletions are formed in mitochondria. However, we can of course not rule out alternative, very low abundant deletion formation caused by other mechanisms. The experimental avenue suggested by the reviewer could be one way to address the existence of such mechanisms, but we are not completely convinced that sequence analysis will provide convincing, conclusive evidence.

In the manuscript, we do not rule out the possible existence of additional mechanisms. However, we would like to point out that this is actually the first study that provides direct, mechanistic evidence for a specific model.

In the manuscript, we have made the following changes to address the reviewer's concern.

First, we have, as requested by reviewer 1, changed “demonstrate” to “suggest” on page 3, last paragraph. The sentence was before: “Our data argues against the previously proposed models and instead demonstrate that mitochondrial deletions are formed by copy choice recombination during active L-strand DNA synthesis.”

We now write:

“Our data argues against the previously proposed models and instead suggest that mitochondrial deletions are formed by copy choice recombination during active L-strand DNA synthesis.”

Second, we have included a number of new experiments to support our model, including next generation sequencing of both *in vivo* and *in vitro* generated mtDNA deletions from different genetic backgrounds, see e.g. figure 5 and supplementary figure 2 in the new version of the manuscript.

Third, we have added a section describing the correlation between *in vivo* and *in vitro* findings in the discussion, page 10, last paragraph:

“In our bioinformatics analysis of breakpoints formed *in vivo*, we identify a subset of imperfect repeats which are used to determine directionality. In the majority of cases, the 5' repeat is retained, but there is also a significant portion for which the opposite is true. Even though we use stringent thresholds in our analysis, we can only predict the most likely breakpoints. The number of incorrect predictions is not known but could be significant. In support of this notion, please consider our *in vitro* reconstitution of deletion formation. In the reconstituted system, we only monitor L-strand synthesis. Therefore, copy-choice recombination can only, in theory, lead to the retention of the repeat close to OriL. However, bioinformatic analysis of deletion products formed *in vitro* generates a directional bias similar to that observed *in vivo*, i.e. in about one third of the cases the 3' repeat is predicted to be retained. We therefore believe that our bioinformatic analysis may underestimate the number of retained 5' repeats and that the actual number may be higher than predicted. In other words, the existence of retained 3'-repeats at 25 – 35% of all breakpoints do not necessarily implicate the existence of alternative mechanisms for deletion formation. In addition, similar nucleotide motifs are enriched at breakpoints both *in vivo* and *in vitro*, adding further support for that the deletion formation process being the same *in vivo* and *in vitro*. In fact, many of the perfect direct repeats >3 bases for which we cannot determine directionality, are shared between deletions formed *in vivo* and *in vitro*.”

#5. The reviewer writes:

Finally, there is one large caveat here – that the deletions were detected in patients with a mutated DNA polymerase. It is possible that the pattern of deletions would be different in other contexts.

Our response:

We agree with the reviewer, this is an important point. We have now analyzed breakpoints in a patient with mtDNA deletions caused by pathogenic variants in the *TOP3A* gene (encoding Topoisomerase 3 α) (Nicholls et al, Mol Cell 2018). In other words, mtDNA deletions formed in the presence of wt POL γ . We applied our computational analyses to reads *in vivo* from the *TOP3A* mutant patient. The results are included in the new version of the manuscript as Supplementary Fig. 2. This mutant background leads to even higher levels of deletions, but the breakpoints identified are similar to those observed with POLG mutants, including the preference for retention of repeats at the 5' side of deletions. We also identified the common deletion in this patient. These data suggest that the proposed model is generally applicable and not only valid in patients with polymerase mutations.

To clarify, our model depends on mtDNA sequences and the specific strand-displacement model. Deletions can take place in the presence of wt POL γ , but will be stimulated by mutations that impair POL γ processivity. Mutations in TOP3A can cause a similar phenotype, most likely due to a topological crisis in the template, which can impair POL γ progression. In other words, many different factors (topology, secondary structures, mutations in POL γ or TWINKLE) can have similar effects, leading to copy-choice recombination.

On page 4, last paragraph, we have added:

“To investigate if the observed directionality was unique to mutations in *POLG*, we also analysed breakpoints in an adult patient with late-onset PEO caused by compound heterozygous mutations in the *TOP3A*, the gene encoding Topoisomerase 3 α (REF), but expressing wild type POL γ . The identified pattern of deletions in the *TOP3A* mutant patient was similar to those associated with *POLG* mutations (Supplementary Fig. 2 and Supplementary Table 1). We identified 399 unique deletions, including the common deletion (frequency > 30). In total, 69 (17.3%) of the identified deletions showed imperfect repeats. Of these, 42 (60.8%) supported retention of the repeat at the 5' side (Supplementary Fig. 2a). Nucleotide motifs enriched at breakpoints were similar to those identified with mutant *POLG* (Supplementary Fig. 2b-c).”

#6. The reviewer writes:

The opening paragraph of the discussion recapitulates the introduction.

Our response

We have omitted a part of this section to avoid recapitulating the same information multiple times.

The omitted section previously read:

“After initiation at OriH, DNA synthesis proceeds in one direction to produce the nascent H-strand. During this process, the parental H-strand is displaced. After about 11,000 bp, the replication machinery reaches OriL, which becomes activated and L-strand DNA synthesis begins.”

REVIEWER #3.

This paper investigates the occurrence and cause of mtDNA deletions. First, mtDNA deletions were identified from muscle samples of three patients with POLG mutations and a POLG disease. Secondly, to recapitulate these deletions, the authors design a substrate that mimics replication of the L-strand synthesis. After replication by wildtype POLG and PCR amplification the authors can identify specific deletion that occur between the direct repeats and potentiated by secondary structures. The deletions identified fit the experimental design, suggesting a similar mechanism described for *E. coli*, copy-choice recombination.

Major comments

There are three major issues with this paper

#1. The reviewer writes:

I doubt there is anything special about POLG as compared to other replicative DNA polymerases that dictate deletion formation and it is likely any polymerase will cause a deletion at direct repeats and secondary structure. It's more about the DNA structure than the polymerase.

Our response:

We completely agree! In fact, this is the most important conclusions of the paper. Previous, unorthodox and rather complicated models to explain mtDNA deletion formation are not necessary. mtDNA deletions are formed following fundamental principles, well accepted and worked out in many other systems. The deletions are mainly due to DNA structures, even if the reactions can be affected by disease causing mutations in e.g. POL γ that influence processivity etc.

The unique patterns of mtDNA deletions (i.e. preferentially in the major arc and preferred directionality) are not due to any unique properties of POL γ or other components of the replication machinery, but due to the strand-displacement mode of replication.

To further clarify this point, we now write on page 7, second to last paragraph:

“Our findings are in agreement with published data which demonstrate that replicative polymerases may cause deletion by copy-choice recombination between direct repeats and that this effect is stimulated by secondary structures. As demonstrated here, this effect can be further accentuated by disease causing mutations in POL γ that impair processivity.”

#2 (part one). The reviewer writes:

There is a disconnect between the in vivo analysis and the in vitro analysis. The in vivo analysis identifies mtDNA deletions in patients with POLG mutations, but no healthy or wt controls are investigated. But the in vitro analysis attempts to recapitulate these deletions with only the WT POLG. Can the authors identify deletions in age matched healthy controls.

Our response:

We agree that healthy controls should be included, and we have now added results from muscle biopsies from two controls (aged 82 and 86 years), presented in Fig. 1B. While comparable in terms of coverage to the *POLG* muscle samples (1500-2000X), very few deletions/duplications were detected in these samples, despite these being older than the *POLG* mutant patients (55, 80 and 61 years) or Top3a patient (67 years). Crucially, these control samples serve the dual

purpose of 1) controlling for false positives that may arise due to technical issues, such as bioinformatics-related aspects (e.g. alignment) or sequencing artefacts, and 2) controlling for the general biological properties of muscle mtDNA, thereby ensuring that the deletions seen in *POLG* mutant samples truly relate to *POLG*.

Additionally, we sequenced mtDNA isolated from blood from two of the patients (*POLG* W748S/R1096C and *POLG* A467T/T251I+P587L). No breakpoints were detected in these samples. Expectedly, mtDNA coverage was lower in these samples compared to muscle DNA. (please see response to reviewer 2, question #1, for additional details).

Regarding the *in vitro* experiments, we have also sequenced the template plasmid, as well those replicated with wild type or muted *POLG*, all at similar coverage (~7800X) and analysed the data using our pipeline. The template samples have a very low abundance of breakpoints compared to the actively replicated wild type and mutant samples for both plasmid constructs built for this study. These results have been included in the manuscript as Fig. 5 and supplementary table 4.

#2 (part two). The reviewer writes:

..can the authors demonstrate enhanced deletion formation *in vitro* with a recombinant *POLG* containing these mutations?

Our response.

We have performed experiments with mutant *POLγ* and observe higher levels of the common deletion. On page 7, we now write:

“We also analyzed deletion formation *in vitro* using equimolar concentrations of two mutated forms of *POLγ*, one with a single amino acid change (A467T) and one with two amino changes in the same allele (T251I+P587L). *In vivo*, these mutated versions of *POLγ*, cause a compound heterozygous form of adPEO²². Interestingly, copy-choice recombination between the two 13-bp repeats were increased with the mutant polymerases (63% compared to 26% for *POLγ* wild type) (Fig. 3d, Supplementary Table 3, Supplementary Fig. 4a-b). Deletions were also observed with T7 DNA polymerase (Supplementary Fig. 5a-d and Supplementary Table 3). No less than 83 % of the deletions formed in the presence of T7 DNA polymerase were due to perfect copy-choice recombination between the two 13-bp repeats (Fig. 3d). Our findings are in agreement with published data demonstrating that replicative polymerases may cause deletion by copy-choice recombination between direct repeats and that this effect is stimulated by secondary structures. As demonstrated here, this effect can be further accentuated by disease causing mutations in *POLγ* that impair processivity.”

#3. The reviewer writes:

It's unclear about certain controls, such as PCR amplification of the T1 substrates without *POLG* and using T7 Pol with thioredoxin.

Our response:

Please see below, response to comment 10.

#4. The reviewer writes:

A normal age matched control for the in vivo deletion analysis was not performed. This analysis needs a healthy control.

Our response:

We have followed the suggestion and added results from two healthy control samples (ages 82 and 86), both sequenced at comparable coverage to the *POLG* mutant samples. Only a small number of breakpoints could be found in these samples (shown in Fig. 1b), despite being older than the *POLG* mutant patients (55, 80 and 61 years) or Top3a patient (67 years).

#5. The reviewer writes:

I don't think there is anything special about *POLG* that contributes to the formation of deletions, which suggests its more about the DNA content and structure that dictates these deletions. To help support that, or not, the authors should investigate other Family A polymerases. A T7 control was included in the supplement, but not compared side by side with *POLG* throughout many of these experiments.

Our response:

Again, we fully agree with the reviewer's statement. There is nothing special with *POLγ* and state that very clearly in the new version of the manuscript.

On page 7, we now write:

"Our findings are in agreement with published data that demonstrate that replicative polymerases may cause deletion by copy-choice recombination between direct repeats and that this effect is stimulated by secondary structures."

To further clarify the similarities between T7 and *POLγ*, we present some data on deletions formed by T7 in figure 3d.

#6. The reviewer writes:

Fig 2d, There needs to be a *POLG* minus control in this experiment to account for deletions generated by the Taq PCR application, especially since Taq is well know to make similar errors in vitro.

Our response:

This is a misunderstanding. We did include a control in figure 2d. Please see lane 8!

We also performed a string of other experiments to address this concern, including the EcoRI restriction experiments presented in Fig 2E and 2F.

In the new version of the manuscript, we have added additional experiments to address these concerns further. We have performed direct next generation sequencing of deletion products formed in vitro, thereby sidestepping the requirement for Taq polymerase (described in the new figure 5).

Finally, we have all along been keenly aware of the problems associated with Taq polymerase and therefore use conditions previously described to minimize deletion formation (Viguera et al. JMB, 2001) as well as template controls.

#7. The reviewer writes:

The *in vitro* analysis focuses on POLG disease patients while the *in vitro* uses only the wild type POLG. Can the authors also test a recombinant POLG *in vitro* with any of these disease mutations to demonstrate enhanced deletion formation.

Our response.

We have performed experiments with mutant POL γ and observe higher levels of the common deletion. On page 7, we now write:

“We also analyzed deletion formation *in vitro* using equimolar concentrations of two mutated forms of POL γ , one with a single amino acid change (A467T) and one with two amino changes in the same allele (T251I+P587L). *In vivo*, these mutated versions of POL γ , cause a compound heterozygous form of adPEO²². Interestingly, copy-choice recombination between the two 13-bp repeats were increased with the mutant polymerases (63% compared to 26% for POL γ wild type) (Fig. 3d, Supplementary Table 3, Supplementary Fig. 4a-b). Deletions were also observed with T7 DNA polymerase (Supplementary Fig. 5a-d and Supplementary Table 3). No less than 83 % of the deletions formed in the presence of T7 DNA polymerase were due to perfect copy-choice recombination between the two 13-bp repeats (Fig. 3d). Our findings are in agreement with published data demonstrating that replicative polymerases may cause deletion by copy-choice recombination between direct repeats and that this effect is stimulated by secondary structures. As demonstrated here, this effect can be further accentuated by disease causing mutations in POL γ that impair processivity.”

#8. The reviewer writes:

Could the authors clarify what is meant by “low” mtDNA SSB levels used in Fig 4B. What is the molecular stoichiometry of the SSB with DNA.

Our response:

We use 100 fmol mtSSB to 10 fmol template. These concentration is well below what is needed to cover the template (mtSSB binds as a tetramer and each tetramer covers a stretch of about 60 nt).

#9. The reviewer writes:

Fig 4b doesn't show any difference between these DNA substrates.

This result is as expected. Please remember that the only difference between the templates is the presence or absence of the 13-bp repeats. All other sequences are the same and at low mtSSB concentrations there are many pause sites. The common deletion is a very rare event and it is therefore not possible to directly observe the heteroduplex during second strand synthesis.

#10. The reviewer writes:

It appears that Supplementary fig 2 with T7 DNA Pol, in which the authors state has a weaker strand-displacing activity, was done in the absence of thioredoxin. If the authors are truly comparing strand displacement activities related to deletion formation, then they should include thioredoxin, similar to POLG2 as a processivity factor.

Our response:

The experiments are done in the presence of thioredoxin (NEB catalog No. M0247). We have not analyzed the strand-displacement activity of T7 DNA polymerase in detail and the comment about a weaker strand-displacing activity was just a suggestion based on the results in supplementary figure 2 (now fig. 3d), which indicated that T7 DNA polymerase did not enter the stem-region. This comment has no relevance for the story and has been removed.

Reviewers' comments:

Reviewer #1 (Remarks to the Author):

The authors did an excellent job in dealing with reviewer comments.

Reviewer #2 (Remarks to the Author):

The authors have performed additional experimental work, but the number of observations remains small (5 individuals, plus the TOP3A case). The new data supports their conclusions, but I still have concerns that the authors make far-reaching conclusions with minimal observations.

They go to great lengths to explain how much care they have put into the bioinformatics, but the detail in the pipeline looks pretty standard. We requested more detail, but it is still not clear what was done. The authors state: "We spent considerable effort testing different tools and parameters while evaluating the results on simulated reads from mitochondrial genomes with in silico generated deletions" - but this is not described to the level that would allow replication of their findings.

It is not clear what the orange dots mean in Fig 1b

Finally, the authors do not seem to recognise the main issue I raised. I will try and explain it again. Given that they are not using a single molecule approach, the detected breakpoints must be present on many molecules. To form these molecules there must be (1) a mutation event and (2) clonal expansion. The repeat structure could influence both of these processes, but the current experimental design cannot separate them. Thus, we cannot be sure that the pattern of breakpoints actually reflects the mutation process, clonal expansion, or both.

Reviewer #3 (Remarks to the Author):

I'm happy to see that the authors have added WT control samples for the in vivo analysis as well as using disease mutant pol gamma enzymes for their in vitro generation of deletions. However, the addition of the mutant enzymes in vitro raises an additional question that I believe the authors can resolve. Specifically, if my understanding is correct, these mutant polymerases, A467T and T251I+P587L are very sick enzymes. So, in the time course of the experiment, how is the polymerase synthesizing products equivalent in length and amount as the wild type (Supp Fig 4A and C, compare lanes 3,4 to 5,6). Is the same amount of enzyme used as the WT. A quick calculation of the WT enzyme used, ~86 ng WT, suggest that a massive and supersaturating amount of enzyme was used for these in vitro experiments. Please explain, why it was necessary to use that much WT enzyme, and how the mutant enzymes performed equivalent to the WT in these reactions.

Further to the question of the in vitro deletion assay by the mutant polymerases, did these reactions generate a higher quantity (frequency) of deletions as the wild type polymerase? Can the authors address the total frequency of deletions made by the mutant as compared to the wild type polymerase?

REVIEWER #2

#1. The reviewer writes:

The authors have performed additional experimental work, but the number of observations remains small (5 individuals, plus the TOP3A case). The new data supports their conclusions, but I still have concerns that the authors make far-reaching conclusions with minimal observations.

Our response:

We have added a sentence to the discussion to address this point. We now write.

“In the current study, we have characterized how mtDNA deletions are formed in five different patients. Future studies of additional patients, will determine if our proposed model can provide a general explanation for deletion formation associated with mitochondrial disease.”

#2. The reviewer writes:

They go to great lengths to explain how much care they have put into the bioinformatics, but the detail in the pipeline looks pretty standard. We requested more detail, but it is still not clear what was done. The authors state: "We spent considerable effort testing different tools and parameters while evaluating the results on simulated reads from mitochondrial genomes with in silico generated deletions" - but this is not described to the level that would allow replication of their findings.

Our response:

Although we expanded on this in the previous revision, we agree that additional details can be added to the methods description, and have elaborated in paragraph two in the relevant section (page 18) to ensure that our approach can be replicated:

Was:

To identify mitochondrial deletions and duplications, alignment to chrM (rCRS assembly) and nuclear chromosomes was initially performed using Bowtie2. Realignment of mitochondrial and unaligned reads to chrM using LAST (E-value < 1e-5) allowed more sensitive identification of gapped alignments indicative of deleted or duplicated segments.

Is now:

To identify mitochondrial deletions and duplications, alignment to chrM (rCRS assembly, NC_012920) and nuclear chromosomes (hg19 assembly) was initially performed using Bowtie2 (parameters: --very-fast). Realignment of mitochondrial and unaligned reads to chrM using LAST (lastdb parameters: -uNEAR; lastal parameters: -Q1 -e80) allowed more sensitive identification of gapped alignments indicative of deleted or duplicated segments. LAST was chosen based on evaluations using simulated sequencing reads, where it was found to show higher accuracy in identification of breakpoints at single bp resolution compared to BLAT and BLAST. Post alignment, we filtered out likely PCR duplicates (mapped at identical positions when taking both mate reads into account, or at the same position in cases where the mate was filtered out in Bowtie2 alignment step). We also disregarded false gapped alignments that may arise in the D-loop region due to the circular mitochondrial genome being represented as a linear sequence.

Additionally, gapped alignments having an *E*-value higher than $1e-5$ or having more than one gap were filtered out.

We agree that the pipeline is, in principle, fairly standard, and at the conceptual level there are probably not so many ways to approach the problem. However, it is our impression that “the devil is in the details” in this case: various small choices in terms of specific tools or parameters used in the pipeline were found to have a big influence on the results. Below we elaborate a bit on the challenges encountered and the choices made when developing the pipeline:

One of the ways we benchmarked the pipeline was by using *in silico* simulated sequencing reads. This enabled us to generate a guaranteed “null” library (lacking deletions or duplication) as well as libraries with breakpoints injected at specific locations. We generated simulated datasets of paired-end sequencing reads from a fasta file representing chrM, chrM with common deletion (8470-13447), and chrM with a duplication (590-1180) (this was done using ART at 2000X coverage).

To identify deletions, we initially mapped the data to chrM using two short read aligners: Subread and BMap. Subread was found to detect the injected deletion/duplication but was also produced fairly large (5-10 bp) errors in the breakpoint positions due to difficulties in handling repeat sequences. Additionally, our “null” simulated dataset revealed major problems with false positive gapped reads with this aligner. BMap, while performing better in some respects, was deemed unsuitable due to an inability to properly handle gapped alignments in cases where 5’/3’ fragments of a gapped read map in the opposite order to the reference sequence. This happens frequently for both deletions and duplications due to the circularity of the mitochondrial genome.

Thus, we could rule out these two options, and instead we zeroed in on three other potential candidates: BLAT, BLASTn and LAST. These were all adequate in terms of false positives, and all performed favourably when identifying the injected common deletion. However, LAST was the most accurate in terms of finding the exact position of the deletion detected:

This can be attributed to sensitive identification of gapped alignments near direct repeats, which is an important prerequisite for the human mitochondrial genome, where structural breakpoints occur preferentially near short direct repeats.

Further, while all three aligners were fairly sensitive when detecting the duplicated region (which had no repeats at the breakpoint), LAST managed to map and identify more reads that spanned the breakpoint compared to BLAT and BLASTn:

LAST thus performed favorably compared to the other four programs we tested and was selected as the aligner of choice.

In our pipeline, before aligning to mtDNA, we are also filtering out possible nuclear contaminants by using an end-to-end fast genomic read aligner (Bowtie2) to map the raw sequencing data on the nuclear genome. This reduces running time and removes reads that may originate from NUMT regions. Here, we made an important observation: some aligners (this may be parameter-dependent) will prioritize an imperfect ungapped alignment to the nuclear genome over a perfect but gapped alignment to the mitochondrial genome. This can lead to massive read dropouts, specifically affecting the very gapped mtDNA reads that are informative of mtDNA deletions. Again, this is a small detail that could have easily been missed, but it has as a major influence on the results.

In addition to evaluations using simulated data, as detailed in the manuscript we also used control samples from healthy individuals to ensure that specificity was high also when using actual sequencing data. In terms of positive training data, we have now applied our pipeline to a fairly wide range of materials, both human and mouse, and have obtained results that are consistent with expectations from orthogonal methodology such as Southern blotting – e.g. see our recently published study in Mol. Cell (Nicholls et al. Jan 2018) where the same pipeline was used. While no method is perfect, we believe that we have in the end established a pipeline that performs very well both in terms of sensitivity and specificity.

#3. The reviewer writes:

It is not clear what the orange dots mean in Fig 1b

Our response:

The reviewer has pointed this out correctly: the orange bars represent the breakpoint frequency at a given position. We have added a description in the figure legend.

#4. The reviewer writes:

Finally, the authors do not seem to recognise the main issue I raised. I will try to explain it again. Given that they are not using a single molecule approach, the detected breakpoints must be present on many molecules. To form these molecules there must be (1) a mutation event and (2) clonal expansion. The repeat structure could influence both of these processes, but the current experimental design cannot separate them. Thus, we cannot be sure that the pattern of breakpoints actually reflects the mutation process, clonal expansion, or both.

Our response:

We agree that repeated generation of the same deletion obviously cannot be separated from clonal expansion of a single deleted parental molecule. However, it must be noted that the deletion patterns we describe in this study generally have a high degree of diversity, in each case involving a spectrum of deletions having a large number of unique combinations of start/end positions. In many of the figures, the total number of unique deletions are indicated, sometimes alongside the total number deletion-supporting fragments (e.g. **Fig. 1e**, **Fig. 5a** and **Fig. 5d**). While the point raised is valid, we do not believe that it has any important bearing on our main conclusions.

In the discussion, we now write:

“It should be noted that our analysis does not allow us to distinguish the repeated generation of the same deletion, from clonal expansion of a single deleted parental molecule. However, we do not believe that this has any important bearing on our main conclusions, since the deletion patterns we describe have a high degree of diversity, in each case involving a spectrum of deletions having a large number of unique combinations of start/end positions.”

REVIEWER #3.

#1. The reviewer writes:

I'm happy to see that the authors have added WT control samples for the in vivo analysis as well as using disease mutant pol gamma enzymes for their in vitro generation of deletions. However, the addition of the mutant enzymes in vitro raises an additional question that I believe the authors can resolve. Specifically, if my understanding is correct, these mutant polymerases, A467T and T251I+P587L are very sick enzymes. So, in the time course of the experiment, how is the polymerase synthesizing products equivalent in length and amount as the wild type (Supp Fig 4A and C, compare lanes 3,4 to 5,6). Is the same amount of enzyme used as the WT. A quick calculation of the WT enzyme used, ~86 ng WT, suggest that a massive and supersaturating amount of enzyme was used for these in vitro experiments. Please explain, why it was necessary to use that much WT enzyme, and how the mutant enzymes performed equivalent to the WT in these reactions.

Our response:

We would prefer not to go into the details of these DNA polymerase gamma mutations, since it would distract from the main message of the manuscript. DNA synthesis in the presence of A467T and T251I/P587L is less efficient than with wild type POLG, with more stalling at multiple locations, as demonstrated by increased smearing and less full-length products in a time-course experiment (see figure 1 for reviewer 2). The precise defects associated with these mutations will

be the subject of a future study, but in essence they do not differ significantly from many other mutant forms of POLG studied by others and us in the past.

Figure 1 for reviewer 2. Time course experiment analyzing L-strand DNA synthesis using the T1 template. Reactions were performed as described in Methods and the experiment is similar to the one described in supplementary figure 4a. Wild type or A467T/T251+P587L mutant POLg were run in parallel in the presence of 500 μ M dNTPs. Time points were 2.5, 5, 7.5, 15, 30, and 60 min.

When we monitor deletion formation *in vitro* with mutant POLG, we use the same concentrations as was used for the wild type POLG (625 fmol). This was not clearly stated in the previous version of the manuscript, but it is corrected in the new version of the materials and methods. The high concentration of wild type POLG was chosen based on a titration experiment, in which we monitored levels of deletions as a function of POLG concentrations (see figure 2 for reviewer 2). As demonstrated, higher concentrations of POLG stimulate deletion formation. We therefore opted for higher levels of POLG to obtain more material to sequence. As is evident from the experiment, deletions are also observed at lower POLG concentrations.

Figure 2 for reviewer 2. Deletion formation during L-strand DNA synthesis at varying POLG concentrations. (Upper panel) L-strand mtDNA synthesis was performed as described in Methods in the presence of the following POLG concentrations: Lane 1, 125 fmol; Lane 2, 250 fmol; Lane 3, 375 fmol; Lane 4, 500 fmol; Lane 5, 625 fmol and Lane 6, 750 fmol. The size markers are linearized dsDNA. (Lower panel) The reactions in the upper panel were analysed by PCR as described in panel the figure legend to figure 2 in the manuscript.

#2. The reviewer writes:

Further to the question of the *in vitro* deletion assay by the mutant polymerases, did these reactions generate a higher quantity (frequency) of deletions as the wild type polymerase? Can the authors address the total frequency of deletions made by the mutant as compared to the wild type polymerase?

Our response:

In the experiments performed here, we use conditions that promote stalling (artificial replication barrier and low mtSSB concentrations). Under these conditions, disease-causing mutations do not cause increased levels of deletion formation compared to wild type POLG. The frequency of deletion formation is indicated in figure 5a. It is difficult to compare the deletion forming capacity of wt and mutant POLG *in vitro*, since mutations that lowers processivity also lowers the overall mtDNA replication levels. However, as noted in figure 3d, we do observe a strong increase of the common deletion with the disease-causing mutations. This effect is discussed in detail on page 6, in the paragraph following the subtitle: “Direct repeats facilitate deletion formation”.

Additional changes not requested by reviewers:

For clarity, we changed the order of the blue/yellow bars in **Fig. 5** to be consistent with the order in the legend and the presentation in **Fig. 1**.

Point-by-point response to the reviewers' comments.

REVIEWER #2

#1. The reviewer writes:

The authors have performed additional experimental work, but the number of observations remains small (5 individuals, plus the TOP3A case). The new data supports their conclusions, but I still have concerns that the authors make far-reaching conclusions with minimal observations.

Our response:

In our work, we have analyzed mtDNA isolated from muscle biopsies from patients with mitochondrial DNA deletion disease. Unfortunately, due to the scarcity of material, we cannot perform a large-scale population study. However, in attempt to address the concerns raised by the reviewer, we have now included additional data in a new figure (Supplementary Figure 3) for one patient with mtDNA deletion disorder caused by a disease-causing mutation in the TWINKLE helicase, i.e. the DNA helicase required at the mitochondrial replication fork. Together with mutations in the POLG gene encoding DNA polymerase gamma, mutations in the TWNK gene encoding the TWINKLE DNA helicase are the most common cause of mtDNA deletions in patients. Similar to the deletion pattern observed in patients with mutations in POLG or TOP3A, the deletion pattern in the patient with a disease-causing mutation in the TWNK gene is consistent with copy-choice recombination.

Page 5, 1st paragraph, we now write:

“We identified fewer mtDNA deletions (180 unique deletions) in the TWNK patient, but one of these was the common deletion (frequency > 11). In total, 26 (14.4%) of the identified deletions showed imperfect repeats. Of these, 15 (58%) supported retention of the repeat at the 5'-side (Supplementary Fig. 3a). Also here the nucleotide motifs enriched at breakpoints were similar to those identified with mutant POLG (Supplementary Fig. 3b-c).”

We would like to point out that the majority of mitochondrial DNA deletion disorders are due to mutations affecting the mitochondrial DNA replication system, which in turn impair DNA replication processivity. Mutations DNA polymerase gamma, TWINKLE, and topoisomerases 3 alpha all lead to problems for the replication machinery to efficiently perform its activity on template DNA. Our report suggests that regardless of the actual mutation in the replication machinery, impaired DNA synthesis causes mtDNA deletions by the same mechanism, copy choice recombination.

Reviewer 2 wrote in the initial assessment that “the key observation is based on deep sequencing three muscle samples and one control cell line”. Here we would like to point out that the clinical sequencing complements the core biochemical studies, which we consider to be the foundation of this study. In our paper, we analyse deletions formed in vivo and in vitro, using the same

bioinformatic pipeline, which allows us to directly compare the results of copy-choice recombination *in vitro*, with the results observed *in vivo*. As demonstrated in the manuscript, we obtain a nearly identical pattern of deletion formation *in vivo* and *in vitro*, strongly supporting our proposed mechanism.

In the previous revision, we added two new control patient samples and two non-muscle control sequencing libraries, plus a sample from a *TOP3A* mutant patient, i.e. non-polymerase mutated. Additionally, as requested by the reviewer we sequenced the *in vitro* samples, adding an additional six sequencing libraries. This collective body of data paints a very consistent picture, with all *POLG* mutant patients as well as the *TOP3A* patient and the *in vitro* data showing similar patterns while all controls are blank. With the addition of a patient with mutations in the *TWINK* gene, we now study mtDNA isolated from five different patients, with disease-causing mutations in three different genes: *POLG* (3 patients), *TWINK* (1 patients) and *TOP3A* (1 patient), which all display the same distinct directionality in deletion formation.

Finally, even if we have added more, supporting *in vivo* data, we acknowledge that we cannot completely rule out that there will be situations where deletions will arise by alternative mechanisms. What these mechanisms would be and how they would work mechanistically are unclear to us, but to address this point, we have added a sentence in the discussion of the manuscript stating in the 2nd paragraph, page 11:

“We have here characterized how mtDNA deletions are formed in five different patients, with mutation in POLG, TOP3A or the TWINK genes. Even if these data support our proposed model, we cannot formally exclude that other mechanisms may also exist. Future studies of additional patients will determine if also other mechanisms may contribute to deletion formation in specific cases.”

#2. The reviewer writes:

They go to great lengths to explain how much care they have put into the bioinformatics, but the detail in the pipeline looks pretty standard. We requested more detail, but it is still not clear what was done. The authors state: "We spent considerable effort testing different tools and parameters while evaluating the results on simulated reads from mitochondrial genomes with *in silico* generated deletions" - but this is not described to the level that would allow replication of their findings.

Our response:

Although we expanded on this in the previous revision, we agree that additional details can be added to the methods description, and have elaborated in paragraph two in the relevant section (page 18) to ensure that our approach can be replicated:

Was:

“To identify mitochondrial deletions and duplications, alignment to chrM (rCRS assembly) and nuclear chromosomes was initially performed using Bowtie2. Realignment of mitochondrial and

unaligned reads to chrM using LAST (E-value < 1e-5) allowed more sensitive identification of gapped alignments indicative of deleted or duplicated segments.”

Is now:

“To identify mitochondrial deletions and duplications, alignment to chrM (rCRS assembly, NC_012920) and nuclear chromosomes (hg19 assembly) was initially performed using Bowtie2 (parameters: --very-fast). Realignment of mitochondrial and unaligned reads to chrM using LAST (lastdb parameters: -uNEAR; lastal parameters: -Q1 -e80) allowed more sensitive identification of gapped alignments indicative of deleted or duplicated segments. LAST was chosen based on evaluations using simulated sequencing reads, where it was found to show higher accuracy in identification of breakpoints at single bp resolution compared to BLAT and BLAST. Post alignment, we filtered out likely PCR duplicates (mapped at identical positions when taking both mate reads into account, or at the same position in cases where the mate was filtered out in Bowtie2 alignment step). We also disregarded false gapped alignments that may arise in the D-loop region due to the circular mitochondrial genome being represented as a linear sequence. Additionally, gapped alignments having an E-value higher than 1e-5 or having more than one gap were filtered out.”

We agree that the pipeline is, in principle, fairly standard, and at the conceptual level there are probably not so many ways to approach the problem. However, it is our impression that “the devil is in the details” in this case: various small choices in terms of specific tools or parameters used in the pipeline were found to have a big influence on the results. Below we elaborate a bit on the challenges encountered and the choices made when developing the pipeline:

One of the ways we benchmarked the pipeline was by using *in silico* simulated sequencing reads. This enabled us to generate a guaranteed “null” library (lacking deletions or duplication) as well as libraries with breakpoints injected at specific locations. We generated simulated datasets of paired-end sequencing reads from a fasta file representing chrM, chrM with common deletion (8470-13447), and chrM with a duplication (590-1180) (this was done using ART at 2000X coverage).

To identify deletions, we initially mapped the data to chrM using two short read aligners: Subread and BMap. Subread was found to detect the injected deletion/duplication but was also produced fairly large (5-10 bp) errors in the breakpoint positions due to difficulties in handling repeat sequences. Additionally, our “null” simulated dataset revealed major problems with false positive gapped reads with this aligner. BMap, while performing better in some respects, was deemed unsuitable due to an inability to properly handle gapped alignments in cases where 5’/3’ fragments of a gapped read map in the opposite order to the reference sequence. This happens frequently for both deletions and duplications due to the circularity of the mitochondrial genome.

Thus, we could rule out these two options, and instead we zeroed in on three other potential candidates: BLAT, BLASTn and LAST. These were all adequate in terms of false positives, and all performed favourably when identifying the injected common deletion. However, LAST was the most accurate in terms of finding the exact position of the deletion detected:

This can be attributed to sensitive identification of gapped alignments near direct repeats, which is an important prerequisite for the human mitochondrial genome, where structural breakpoints occur preferentially near short direct repeats.

Further, while all three aligners were fairly sensitive when detecting the duplicated region (which had no repeats at the breakpoint), LAST managed to map and identify more reads that spanned the breakpoint compared to BLAT and BLASTn:

LAST thus performed favorably compared to the other four programs we tested and was selected as the aligner of choice.

In our pipeline, before aligning to mtDNA, we are also filtering out possible nuclear contaminants by using an end-to-end fast genomic read aligner (Bowtie2) to map the raw sequencing data on the nuclear genome. This reduces running time and removes reads that may originate from NUMT regions. Here, we made an important observation: some aligners (this may be parameter-dependent) will prioritize an imperfect ungapped alignment to the nuclear genome over a perfect but gapped alignment to the mitochondrial genome. This can lead to massive read dropouts, specifically affecting the very gapped mtDNA reads that are informative of mtDNA deletions. Again, this is a small detail that could have easily been missed, but it has as a major influence on the results.

In addition to evaluations using simulated data, as detailed in the manuscript we also used control samples from healthy individuals to ensure that specificity was high also when using actual sequencing data. In terms of positive training data, we have now applied our pipeline to a

fairly wide range of materials, both human and mouse, and have obtained results that are consistent with expectations from orthogonal methodology such as Southern blotting – e.g. see our recently published study in Mol. Cell (Nicholls et al. Jan 2018) where the same pipeline was used. While no method is perfect, we believe that we have in the end established a pipeline that performs very well both in terms of sensitivity and specificity.

#3. The reviewer writes:

It is not clear what the orange dots mean in Fig 1b

Our response:

The reviewer has pointed this out correctly: the orange bars represent the breakpoint frequency at a given position. We have added a description in the figure legend.

#4. The reviewer writes:

Finally, the authors do not seem to recognise the main issue I raised. I will try and explain it again. Given that they are not using a single molecule approach, the detected breakpoints must be present on many molecules. To form these molecules there must be (1) a mutation event and (2) clonal expansion. The repeat structure could influence both of these processes, but the current experimental design cannot separate them. Thus, we cannot be sure that the pattern of breakpoints actually reflects the mutation process, clonal expansion, or both.

Our response:

We agree that repeated generation of the same deletion obviously cannot be separated from clonal expansion of a single deleted parental molecule. However, it must be noted that the deletion patterns we describe in this study generally have a high degree of diversity, in each case involving a spectrum of deletions having a large number of unique combinations of start/end positions. In many of the figures, the total number of unique deletions are indicated, sometimes alongside the total number deletion-supporting fragments (e.g. **Fig. 1e**, **Fig. 5a** and **Fig. 5d**). While the point raised is valid, we do not believe that it has any important bearing on our main conclusions.

In the discussion, 1st paragraph, page 11, we now write:

“It should be noted that our analysis does not allow us to distinguish the repeated generation of the same deletion, from clonal expansion of a single deleted parental molecule. However, we do not believe that this has any important bearing on our main conclusions, since the deletion patterns we describe have a high degree of diversity, in each case involving a spectrum of deletions having a large number of unique combinations of start/end positions.”

REVIEWER #3.

#1. The reviewer writes:

I'm happy to see that the authors have added WT control samples for the in vivo analysis as well as using disease mutant pol gamma enzymes for their in vitro generation of deletions. However, the addition of the mutant enzymes in vitro raises an additional question that I believe the authors can resolve. Specifically, if my understanding is correct, these mutant polymerases, A467T and

T251I+P587L are very sick enzymes. So, in the time course of the experiment, how is the polymerase synthesizing products equivalent in length and amount as the wild type (Supp Fig 4A and C, compare lanes 3,4 to 5,6). Is the same amount of enzyme used as the WT. A quick calculation of the WT enzyme used, ~86 ng WT, suggest that a massive and supersaturating amount of enzyme was used for these *in vitro* experiments. Please explain, why it was necessary to use that much WT enzyme, and how the mutant enzymes performed equivalent to the WT in these reactions.

Our response:

We would prefer not to go into the details of these DNA polymerase gamma mutations, since it would distract from the main message of the manuscript. DNA synthesis in the presence of A467T and T251I/P587L is less efficient than with wild type POLG, with more stalling at multiple locations, as demonstrated by increased smearing and less full-length products in a time-course experiment (see figure 1 for reviewer 2). The precise defects associated with these mutations will be the subject of a future study, but in essence they do not differ significantly from many other mutant forms of POLG studied by others and us in the past.

Figure 1 for reviewer 2. Time course experiment analyzing L-strand DNA synthesis using the T1 template. Reactions were performed as described in Methods and the experiment is similar to the one described in supplementary figure 4a. Wild type or A467T/T251I+P587L mutant POLg were run in parallel in the presence of 500 μ m dNTPs. Time points were 2.5, 5, 7.5, 15, 30, and 60 min.

When we monitor deletion formation *in vitro* with mutant POLG, we use the same concentrations as was used for the wild type POLG (625 fmol). This was not clearly stated in the previous version of the manuscript, but it is corrected in the new version of the materials and methods. The high concentration of wild type POLG was chosen based on a titration experiment, in which we monitored levels of deletions as a function of POLG concentrations (see figure 2 for reviewer 2). As demonstrated, higher concentrations of POLG stimulate deletion formation. We

therefore opted for higher levels of POLG to obtain more material to sequence. As is evident from the experiment, deletions are also observed at lower POLG concentrations.

Figure 2 for reviewer 2. Deletion formation during L-strand DNA synthesis at varying POLG concentrations. (Upper panel) L-strand mtDNA synthesis was performed as described in Methods in the presence of the following POLG concentrations: Lane 1, 125 fmol; Lane 2, 250 fmol; Lane 3, 375 fmol; Lane 4, 500 fmol; Lane 5, 625 fmol and Lane 6, 750 fmol. The size markers are linearized dsDNA. (Lower panel) The reactions in the upper panel were analysed by PCR as described in panel the figure legend to figure 2 in the manuscript.

#2. The reviewer writes:

Further to the question of the *in vitro* deletion assay by the mutant polymerases, did these reactions generate a higher quantity (frequency) of deletions as the wild type polymerase? Can the authors address the total frequency of deletions made by the mutant as compared to the wild type polymerase?

Our response:

In the experiments performed here, we use conditions that promote stalling (artificial replication barrier and low mtSSB concentrations). Under these conditions, disease-causing mutations do not cause increased levels of deletion formation compared to wild type POLG. The frequency of deletion formation is indicated in figure 5a. It is difficult to compare the deletion forming capacity of wt and mutant POLG *in vitro*, since mutations that lowers processivity also lowers the overall mtDNA replication levels. However, as noted in figure 3d, we do observe a strong increase of the common deletion with the disease-causing mutations. This effect is discussed in

detail on page 6, in the paragraph following the subtitle: “Direct repeats facilitate deletion formation”.

Additional changes not requested by reviewers:

For clarity, we changed the order of the blue/yellow bars in **Fig. 5** to be consistent with the order in the legend and the presentation in **Fig. 1**.

Reviewers' comments:

Reviewer #2 (Remarks to the Author):

The authors have attempted to respond to my ongoing concerns, but there is no new data, not any new explanations to definitively address the same concerns I raised in both previous reviews. I am afraid it is a little irritating to be reminded that there is 'scarcity of material'. The authors have built careers and an international centre based on large patient cohorts with associated muscle biopsy material – but they have only studied a handful here. I have never suggested that the authors carry out a population-based study, which would be ridiculous. I was simply asking for them to validate their key findings on a few independent samples for each of the nuclear genetic defects causing the multiple deletions. I do not think that replicating the key findings in independent samples is unreasonable. I can see that the other reviewer is also interested in this level of detail. For example, the controls and patients were not sequenced at the same time or in the same batch, and there were marked differences in coverage. Validating their findings independently on more patient samples would address these concerns, and others.

It is also a little annoying that we have had to extract the detailed bioinformatic methodology through two rounds of peer review. I totally agree with the authors that the 'devil is in the detail' – this is precisely why I requested details in the first review (which were not forthcoming). I now see that further detail is required. Can the authors show objectively why LAST performed more effectively? Did they carry out a statistical analysis to validate this choice? The data presented in the response (upper figure) seems to show BLAT is best (although they are all very similar). The data certainly doesn't show that LAST is 'the most accurate'.

They then go on to say that their pipeline (which includes removing NUMTs – which is fairly standard), has a 'major influence on the results'. This is exactly my point – can they show that their pipeline is superior to other pipelines? This would convince me that the bioinformatic advances they report are really the key to their findings, and it would also allow others to replicate their findings. They 'believe that their pipeline performs well in terms of sensitivity and specificity' but they simply do not show this. It would be possible to do this with spiked samples using PCR amplicons, or well-characterised genomic DNA. The field of CNV detection (including mtDNA deletions) is known to be very challenging, and they present their method as a solution to this. It is not unreasonable to expect to see this clearly demonstrated, and the linked code to be made available for the research community.

With regards the final point, the authors argue that the diversity of different deletions that they report favours their mechanistic hypothesis. Surely they mean that there is a diversity of deletions, but with a common theme pointing towards a unifying mechanism? My point is that the common theme could be due to clonal expansion.....not de novo deletion formation. This important caveat needs to be very prominent in the discussion so as not to mislead the readers.

Reviewer #3 (Remarks to the Author):

The revision is much improved and I only have one minor item that needs clarification.

Can the authors please define "frequency" at the top of page 5 for the common deletion, line 127 and line 132, referring to the frequency of the common deletion, >30 and >11. Does this refer to the actual number of occurrences or is it a percentage. Why is it ">" and not an absolute number?

Re. NCOMMS-17-31198C-Z

Point-by-point response to the reviewer's comments.

Reviewer #2 (Remarks to the Author):

The reviewer writes:

"The authors have attempted to respond to my ongoing concerns, but there is no new data, not any new explanations to definitively address the same concerns I raised in both previous reviews. I am afraid it is a little irritating to be reminded that there is 'scarcity of material'. The authors have built careers and an international centre based on large patient cohorts with associated muscle biopsy material – but they have only studied a handful here. I have never suggested that the authors carry out a population-based study, which would be ridiculous. I was simply asking for them to validate their key findings on a few independent samples for each of the nuclear genetic defects causing the multiple deletions. I do not think that replicating the key findings in independent samples is unreasonable. I can see that the other reviewer is also interested in this level of detail. For example, the controls and patients were not sequenced at the same time or in the same batch, and there were marked differences in coverage. Validating their findings independently on more patient samples would address these concerns, and others."

Our response:

We are a bit confused by this comment, since there is new data. The reviewer seems to have failed to notice that we did indeed add more patient data in the last revision (NCOMMS-17-31198C-Z), specifically a pathogenic mutation in the *TWINK* helicase, thus expanding the patient repertoire with yet another type of replication defect. Furthermore, in the previous revision round, data from a *TOP3A* mutant patient was added in response to comments from the same reviewer, in addition to the three independent *POLG* patients that were included already in the first submission. We have thus added more patient-data, both during the first and second round of revision. Furthermore, we also added two control patient samples and two non-muscle control sequencing libraries in the first revision, specifically in response to demands from the same reviewer. Additionally, again as requested by reviewer 2, we sequenced all the *in vitro* samples, adding an additional six libraries.

We have thus taken the concerns raised by reviewer 2 very seriously and are a bit taken back by the statement that the reviewer is getting irritated with us, when we have really tried to address his/her concerns. We have spent considerable efforts and also used very valuable and scarce patient material. In addition, we would like to point out that the entire process would have been simpler, if the reviewer had been clear about this point in the first round of critique of the paper. In his/her original comments, the reviewer wrote:

"Finally, there is one large caveat here – that the deletions were detected in patients with a mutated DNA polymerase. It is possible that the pattern of deletions would be different in other contexts."

We believe that we have addressed this concern, by adding two more patients with different disease-causing mutations (in *TWINK* and *TOP3A*). It is worth noticing that mutations in *TWINK* and *POLG* are the major causes of mtDNA deletions. However, this appears not to be enough for the reviewer, since in the latest round of review he/she writes:

"I was simply asking for them to validate their key findings on a few independent samples for each of the nuclear genetic defects causing the multiple deletions."

We have indeed addressed the original concern raised by the reviewer. It is difficult for us to continue adding more data, based on new demands not clearly stated in the original review of the paper. It feels a bit like trying to hit a moving target.

Regarding the reviewer's comment: *"For example, the controls and patients were not sequenced at the same time or in the same batch, and there were marked differences in coverage"*, the controls were added in response to earlier requests from this reviewer, and it is thus not surprising they were not sequenced in the same batch. There is however little reason to believe that batch effects could have any major influence on the kind of deletion patterns that are being detected, which notably are consistent with what has been described previously (for instance, Samuels et al, 2004; Chen et al, 2011). Low chrM coverage in blood samples is expected given the low mitochondrial DNA copy number in these cells. The muscle patient muscle samples have somewhat higher coverage (2229x, 2218x and 1127x; on average 1858x) compared to the muscle controls (846x and 1692x; on average 1269x), but this is irrelevant: both groups have ample coverage and the differences in the results comparing patients and controls are massive and not explainable by differential coverage. There are low levels of deletions also in the healthy controls (82 and 86 years old), which is expected, since deletions accumulate in postmitotic tissue with age (Bender et al, 2006). To further alleviate any concerns about this, we subsampled the two higher coverage patient samples to such that half of the reads were randomly removed, making patients and controls comparable in depth (Figure 1 for review). As shown below, patient samples remain clearly distinct from the healthy controls. Please note that the healthy controls have some deletions, which is expected due to their high age.

Figure 1 for review.

The reviewer writes:

"It is also a little annoying that we have had to extract the detailed bioinformatic methodology through two rounds of peer review. I totally agree with the authors that the 'devil is in the detail' – this is precisely why I requested details in the first review (which were not forthcoming). I now see that further detail is required. Can the authors show objectively why LAST performed more effectively? Did they carry out a statistical analysis to validate this choice? The data presented in the response (upper figure) seems to show BLAT is best (although they are all very similar). The data certainly doesn't show that LAST is 'the most accurate'.

They then go on to say that their pipeline (which includes removing NUMTs – which is fairly standard), has a 'major influence on the results'. This is exactly my point – can they show that their pipeline is superior to other pipelines? This would convince me that the bioinformatic advances they report are really the key to their findings, and it would also allow others to replicate their findings. They 'believe that their pipeline performs well in terms of sensitivity and specificity' but they simply do not show this. It would be possible to do this with spiked samples using PCR amplicons, or well-characterised genomic DNA. The field of CNV detection (including mtDNA deletions) is known to be very challenging, and they present their method as a solution to this. It is not unreasonable to expect to see this clearly demonstrated, and the linked code to be made available for the research community."

Our response:

The current methods text to our knowledge contains all the details needed to implement the pipeline and in the previous point by point we described in great detail many of the choices and considerations made when implementing the pipeline. We have not argued that our method is superior to other pipelines and, importantly, we do not believe that the results in this study hinge on the pipeline being implemented exactly as done in this case.

The text accompanying the first figure highlights the fact that LAST was most accurate in pinpointing the correct coordinates for the injected deletion, but we agree that this could have been better described. However, we want to make it clear that we believe that LAST and BLAT are both sensitive aligners for detection of large structural alterations in mitochondrial genomes. LAST was chosen by weighting several factors together. In addition to what was said in the previous response, it has documented good performance in repetitive regions and unlike BLAT it considers FASTQ quality scores during alignment (Frith et al., 2010; Frith, 2011). Additionally, it is multithreaded and runs fairly fast. We also want to stress that evaluations on simulated datasets (downloadable at <http://bit.ly/mtsim>) made it clear some other aligners that we tried were outright unsuitable for the task, as described in our previous response.

As requested by the reviewer, we have made our pipeline available for download, with a reference added to Methods (Mitopoint1.0: http://bit.ly/mipt_v1). Additionally, based on above mentioned simulated datasets, we have compared our pipeline with a recently published workflow (MitoDel3.0; Bosworth et al., 2017) which relies on BLAT to identify deletions. Both tools report essentially no false positives, but as can be seen below, the fraction of injected deletion-or duplication-supporting reads detected with our pipeline is considerably higher:

Thus, even though the authors have chosen an aligner that in principle is well suited for the task, and despite the overall principle of the implementation is the same as ours (as noted previously there are not so many ways of doing this), the performance differs considerably. Specifically, the way in which reads are extracted from the initial .bam file, after a first run using a short read aligner, as well as later parsing of the BLAT file seems to influence the performance negatively, resulting in a high percentage of false negatives.

The reviewer writes:

“With regards the final point, the authors argue that the diversity of different deletions that they report favours their mechanistic hypothesis. Surely they mean that there is a diversity of deletions, but with a common theme pointing towards a unifying mechanism? My point is that the common theme could be due to clonal expansion.....not de novo deletion formation. This important caveat needs to be very prominent in the discussion so as not to mislead the readers.”

Our response:

We took this comment very seriously and gave an extensive response to this point in the previous revision. We are a bit surprised that the question comes up again. Has the reviewer really noticed our previous response? Or could it be that he/she has been looking at a previous revision?

In the previous round of revision the reviewer asked:

“Finally, the authors do not seem to recognise the main issue I raised. I will try an explain it again. Given that they are not using a single molecule approach. the detect breakpoints must be present on many molecules. To form these molecules there must be (1) a mutation event and (2) clonal expansion. The repeat structure could influence both of these processes, but the current experimental design cannot separate them. Thus, we cannot be sure that the pattern of breakpoints actually reflects the mutation process, clonal expansion, or both.”

Our answer to this question was as follows:

We agree that repeated generation of the same deletion obviously cannot be separated from clonal expansion of a single deleted parental molecule. However, it must be noted that the deletion patterns we describe in this study generally have a high degree of diversity, in each case involving a spectrum of deletions having a large number of unique combinations of start/end positions. In many of the figures, the total number of unique deletions are indicated, sometimes alongside the total number deletion-supporting

fragments (e.g. **Fig. 1e**, **Fig. 5a** and **Fig. 5d**). While the pointed raised is valid, we do not believe that it has any important bearing on our main conclusions.

As requested by the reviewer, we also added the following section in the discussion (1st paragraph, page 11).

“It should be noted that our analysis does not allow us to distinguish the repeated generation of the same deletion, from clonal expansion of a single deleted parental molecule. However, we do not believe that this has any important bearing on our main conclusions, since the deletion patterns we describe have a high degree of diversity, in each case involving a spectrum of deletions having a large number of unique combinations of start/end positions.”

Reviewer #3 (Remarks to the Author):

The reviewer writes:

The revision is much improved and I only have one minor item that needs clarification.

Can the authors please define “frequency” at the top of page 5 for the common deletion, line 127 and line 132, referring to the frequency of the common deletion, >30 and >11. Does this refer to the actual number of occurrences or is it a percentage. Why is it “>” and not an absolute number?

Our response:

Thank you for pointing this out! We have corrected the text to clarify that this refers to actual number of occurrences.

References

Bender, A., Krishnan, K.J., Morris, C.M., Taylor, G.A., Teeve, A.K., Perry, R.H., Jaros, E., Hersheson, J.S., Betts, J., Klopstock, T., Taylor R.W. and Turnbull, D.M (2006) High levels of mitochondrial DNA deletions in substantia nigra neurons in aging and Parkinson disease. *Nature Genet.* 38, 515-7

Bosworth, C.M., Grandhi, S., Gould, M.P., and Chen, T., He, J., and Zhao, W., (2011) The generation of mitochondrial DNA large-scale deletions in humans, *Journal of Human Genetics* 56, 689-94

Frith, M.C. (2011). Gentle Masking of Low-Complexity Sequences Improves Homology Search. *PLOS ONE* 6, e28819.

Frith, M.C., Wan, R., and Horton, P. (2010). Incorporating sequence quality data into alignment improves DNA read mapping. *Nucleic Acids Res.* 38, e100–e100.

LaFramboise, T. (2017). Detection and quantification of mitochondrial DNA deletions from next-generation sequence data. *BMC Bioinformatics* 18, 407.

Samuels, D.C., Schon, E.A. and Chinnery, P.F. (2004) Two direct repeats cause most human mtDNA deletions. *Trends Genet* **20**, 393-8

Reviewers' comments:

Reviewer #2 (Remarks to the Author):

The bioinformatic pipeline is now well described, albeit that it is not particularly special, nor a significant conceptual/technical advance as was suggested in earlier submissions.

One minor issue does remain - the depth of the sequencing used in cases and controls was different, and this could bias the interpretation. Can the authors address this by random sampling from the higher depth dataset, to make the approaches more comparable?

Re. NCOMMS-17-31198D

Response to the reviewer's comment.

Reviewer #2 (Remarks to the Author):

One minor issue does remain - the depth of the sequencing used in cases and controls was different, and this could bias the interpretation. Can the authors address this by random sampling from the higher depth dataset, to make the approaches more comparable?

Our response:

To alleviate the concern, we subsampled the two higher coverage patient samples to such that half of the reads were randomly removed, making patients and controls comparable in depth and included the results as Supplementary Fig 2 in the manuscript. Expectedly, patient samples remain clearly distinct from the healthy controls. Please note that the healthy controls have some deletions, which is expected due to their high age (82 and 86 years of age).

Page 4, 1st paragraph, we now write:

Deletions could be detected also in the healthy controls, which was as expected due to the high age of these individuals (82 and 86 years). The levels of deletions in the patient samples were however much higher, even after correcting for variable sequencing depth (Supplementary Fig. 2).